# TRAINING LANGUAGE MODELS ON SYNTHETIC EDIT SEQUENCES IMPROVES CODE SYNTHESIS

**Ulyana Piterbarg, Lerrel Pinto, & Rob Fergus**[*]
New York University

## ABSTRACT

Software engineers mainly write code by editing existing programs. In contrast, language models (LMs) autoregressively synthesize programs in a single pass. One explanation for this is the scarcity of sequential edit data. While high-quality instruction data for code synthesis is scarce, edit data for synthesis is even scarcer. To fill this gap, we develop a synthetic data generation algorithm called LintSeq. This algorithm refactors programs into sequences of synthetic edits by using a *linter* to procedurally sample across interdependent lines of source code. Synthetic edits sampled with LintSeq reflect the syntax and semantics of their programming language. To test the algorithm, we use it to refactor a dataset of instruction + program pairs into instruction + program-diff-sequence tuples. Then, we fine-tune a series of smaller LMs ranging from 2.6B to 14B parameters on both the re-factored and original versions of this dataset. We perform comprehensive evaluations comparing edit sequence code LMs against baselines on HumanEval, MBPP(+), CodeContests, DS-1000, and BigCodeBench. We show that models fine-tuned to iteratively synthesize code match or outperform baselines on pass@1, and exhibit better scaling across higher pass@k as a function of total test-time FLOPs. Finally, we also pretrain our own tiny LMs for code understanding. We show that fine-tuning these models to synthesize code edit-by-edit results in strong performance on HumanEval and MBPP(+) compared to existing code language models of similar scale such as CodeT5+, AlphaCode, and Codex.

## 1 INTRODUCTION

The successes of language models (LMs) are difficult to overstate. However, consistent and correct zero-shot generation in code synthesis remains out-of-reach for all but the largest models (Abdin et al., 2024; Groeneveld et al., 2024; Dubey et al., 2024). Compared to other reasoning tasks, this setting has two challenging properties, namely solutions are both long and structured.

Humans tackle problems that have these properties by leveraging abstract mental models, first developing a plan for their solution that reflects the setting's structure and then executing the plan one step at a time (Gopnik, 1982; Kirsh, 2009). For example, a software engineer might employ object-oriented programming when creating a new code-base by developing a "class" object and then gradually adding new functionality to this class as their code-base becomes more complex.

In contrast, LMs are trained to autoregressively synthesize entire programs from scratch. This makes repeatedly editing a program with an LM extremely expensive – current state-of-the-art, LM-powered code editing tools like Cursor repeatedly prompt models to rewrite entire programs during every edit generation call (Sanger, 2024). LM outputs also suffer from degrading quality as sequence lengths grow and exhibit limited diversity across samples (Chen et al., 2021; Li et al., 2022b; Roziere et al., 2023; Lozhkov et al., 2024). The consequence of these pathologies is that there does not exist a reliable trade-off between zero-shot generation quality and total test-time compute under the current paradigm of autoregressive code synthesis, particularly for smaller LMs.

In this paper, we claim that these issues can be mitigated at the data-level by reparameterizing code synthesis as a sequential edit problem. Rather than training models for single-step generation of entire

---

[*]We open-source our code and models to `https://lintseq.github.io/`. **Contact:** {`up2021, lerrel, fergus`}@cs.nyu.edu.

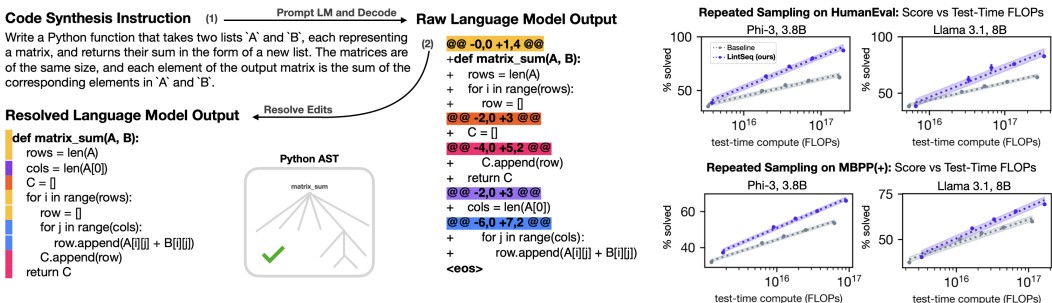

Figure 1: **Code synthesis with LMs trained on synthetic code edit sequences**. Left: An example generation from an LM trained to synthesize code as a stream of linter-error-free edits. Right: **Training LMs to write code edit-by-edit by preprocessing instruction data for SFT with LintSeq improves test-time scaling laws during repeated sampling**, i.e. the percentage of benchmark problems solved by any attempt (pass@k) as a function of total test-time FLOPs compared to training on standard data (see Appendix A.4). Shading indicates standard error in linear fit.

programs, we propose that models be trained to generate code *edit-by-edit*. This objective has a major obstacle: while datasets of filtered GitHub repository commits like CommitPackFT (Muennighoff et al., 2023) have dramatically improved the quality of open-source code edit data, they contain limited sequential data. Moreover, the edits in such these datasets reflect the granularity at which programmers save code, but not necessarily the granularity at which they write and/or reason about it.

To address this, we introduce a sampling algorithm called "LintSeq" that can be used to express any program in a training corpus as a sequence of structured code edits. LintSeq leverages linters – simple code analysis tools that check programs for errors and stylistic issues – to ensure that each generated edit meaningfully reflects the syntactical structure of the programming language that it is written in. The algorithm consists of two phases: a backward phase, which takes a source file as input and samples code deletions from this file to yield possible sequences of *linter-error-free* intermediate program states; and a forward edit computation phase, which reverses each sampled program sequence, employs the Unix `diff` (Thompson & Ritchie, 1975) operator to compute deltas between consecutive versions of each file, and outputs the generated edit sequences. LMs trained on data sampled with LintSeq synthesize code by repeatedly predicting insertion edits to files.

To test the impact of training LMs on synthetic edit sequences sampled with LintSeq, we conduct a series of supervised fine-tuning (SFT) experiments. In each experiment, we compare the performance of models trained on a corpus of example programs re-sampled into synthetic edit sequences with LintSeq to those trained on the original dataset. We evaluate LMs zero-shot and without chain-of-thought on HumanEval (Chen et al., 2021), MBPP (Austin et al., 2021), DS-1000 (Lai et al., 2023), BigCodeBench (Zhuo et al., 2024), and CodeContests (Li et al., 2022b) on "pass@k," the proportion of problems solved by any attempt given "k" tries. Our results show the following:

1. Across models ranging in scale from 150M to 14B parameters, training LMs to iteratively synthesize programs improves the diversity of model-generated code compared to training on standard instruction data, while either preserving or improving code quality.

2. The improved diversity of generated programs means that pass@k performance increases faster as a function of test-time FLOPs, allowing for a better trade-off between the two.

3. Ablating the linter from edit sampling during data generation hurts the downstream quality of programs synthesized by edit sequence models.

## 2 LINTSEQ: CODE SYNTHESIS AS A SEQUENTIAL EDIT PROBLEM

The key to solving a hard problem often lies in knowing how to decompose it into sub-problems. LintSeq is an algorithm for synthetic data generation that decomposes programs in training corpuses across insertion edit chunks that reflect the syntax and semantics of their programming language. To sample such chunks, it uses a code linter. The algorithm is inspired by recent work on discrete diffusion methods for text generation, where decoding is non-autoregressive (Li et al., 2022a).

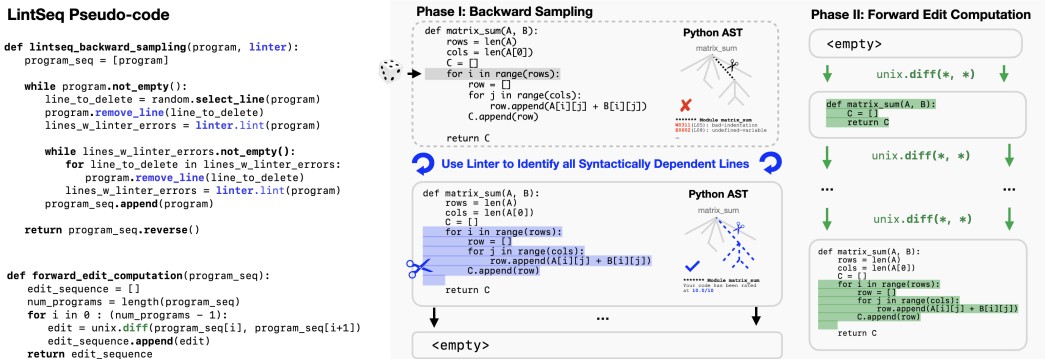

Figure 2: **LintSeq: Training LMs to write code edit-by-edit with supervised learning by generating synthetic data**. LintSeq decomposes existing programs into synthetic edits that reflect the syntax & semantics of their programming language. At each iteration, the algorithm samples an edit chunk from a program by: randomly selecting a line of code to delete; identifying the minimal set of lines that are dependent on this line with a code linter; and finally, removing the line and its dependents. These steps are repeated until all lines of code have been removed. LintSeq then processes the reversed sequence of program states with Unix-diff to express it as a sequence of edits.

Informally, the hypothesis underlying LintSeq is as follows: by training LMs to synthesize code edit-by-edit on large-scale datasets, we can potentially achieve a better trade-off between generation quality and test-time compute while still benefiting from the training and sampling efficiency of autoregressive language modeling. In this section, we define important terms, provide a formalism for the edit sequence re-parameterization of code synthesis, and formally introduce LintSeq.

## 2.1 DEFINITIONS

We define a *linter* to be a static code analysis tool that scans source code for defects. Linters can identify code that is objectively incorrect, throwing errors if and when a program contains syntax errors or refers to non-existent variables or packages. It is important to note that unlike a formal verifier, linters may return false positives, i.e. they may be unable to detect more complex errors, particularly in dynamically typed programming languages like Python or JavaScript.

For a given source file, define an *intermediate program state* to be a program that contains only a subset of the line-by-line contents of the original file, such that the order of these lines is preserved. We call an intermediate program state *linter-error-free* if checking this program with an appropriate linter produces exactly the same error trace(s) as those output when checking the original source file.

## 2.2 REPRESENTING CODE WITH EDIT SEQUENCES

We operate in the textual supervised learning setting in this paper, where we have access to a code dataset $\mathcal{D}$ of $N$ example programs $y$, each of which may be optionally paired with a corresponding natural language instruction $x$ that describes the program's function, i.e. $\mathcal{D} = \{(x^i, y^i)\}_{i=1}^N$.

Let $\Delta(\cdot, \cdot)$ denote the Unix `diff` operator (Thompson & Ritchie, 1975), which computes a text difference between a pair of strings by performing a line-by-line matching and returns a summary of the detected differences. The `diff` operator is implemented by popular version control and development systems to help programmers track edits between versions of text files. A single edit computed with the `diff` operator may consist of multiple line deletions and/or line insertions.

Fix a program $y$ in the dataset $\mathcal{D}$. Consider a sequence of $\boldsymbol{\sigma}_y$ of $j$ text strings corresponding to programs that terminates at $y$, $\boldsymbol{\sigma}_y = (y_1, \ldots, y_{j-1}, y)$. We can equivalently re-express $\boldsymbol{\sigma}_y$ as an edit sequence $\boldsymbol{\delta}_y$ of length $j$ by first computing a diff between an empty program $\varepsilon$ and the first program in the sequence, and then computing diffs between all pairs of consecutive programs, as shown below.

$$\boldsymbol{\delta}_y = (\Delta(\varepsilon, y_1), \Delta(y_1, y_2), \Delta(y_2, y_3), \ldots, \Delta(y_{j-1}, y)) \tag{1}$$

If $\mathcal{D}'$ is a dataset such that for every pair $(x, y) \in \mathcal{D}$, there exists a pair $(x, \boldsymbol{\delta}_y) \in \mathcal{D}'$, then we say that $\mathcal{D}'$ is an *edit sequence refactoring* of $\mathcal{D}$.

## 2.3 GENERATING LINTER-GUIDED SYNTHETIC EDIT SEQUENCES

Recall from above that a single program edit computed by the `diff` operator $\Delta(\cdot, \cdot)$ can consist of any number of deletions and insertions. LintSeq is an algorithm for computing edit sequence refactorings $\mathcal{D}'$ such that all data $(x, \boldsymbol{\delta}_y) \in \mathcal{D}'$ have a particular property: every edit in $\boldsymbol{\delta}_y$ consists of *insertions only*. There are two phases in LintSeq: a backward sampling phase that is used to compute program state sequences $\boldsymbol{\sigma}_y$, and a forward edit sequence computation phase that is used to re-express $\boldsymbol{\sigma}_y$ as edit sequences $\boldsymbol{\delta}_y$. Pseudo-code as well as a visualization of each of these phases is provided in Figure 2. Full examples of edit sequences generated with LintSeq are provided in Appendix F (Figures 9 and 10).

**Phase I: Backward Sampling** In the backward sampling phase of LintSeq, for each of the $N$ pairs $(x, y) \in \mathcal{D}$, we generate $s$ sequences of intermediate program states $\boldsymbol{\sigma}_y$ that begin with the empty program and terminate at the original program $y$. These sequences are generated in reverse or backwards using a simple procedure that we dub *linter-guided sampling*. Starting with the program $y$, we sequentially generate each predecessor program in $\boldsymbol{\sigma}_y$ from its successor by following these steps: (1) delete a line from the current program by sampling uniformly at random; (2) run a linter or other verifier on the remaining code; (3) if the deletion induced new errors, remove all affected lines; and (4) repeat steps 2 and 3 until no errors are caught by the linter. We repeat these steps until all lines have been removed from the original program $y$, at which point $\boldsymbol{\sigma}_y$ has been generated.

**Phase II: Forward Edit Computation** Once $s$ program state sequences $\boldsymbol{\sigma}_y$ have been generated for each $(x, y) \in \mathcal{D}$, we run the forward edit computation phase of our algorithm. In this phase, we apply Equation 1 from above to compute an edit sequence $\boldsymbol{\delta}_y$ for each $\boldsymbol{\sigma}_y$. Starting from the last program that was added to $\boldsymbol{\sigma}_y$, we use the `diff` operator to compute edits between each pair of consecutive programs in $\boldsymbol{\sigma}_y$ up to the original program $y$. Finally, we pair each edit sequence $\boldsymbol{\delta}_y$ with its instruction $x$ (if present) to yield an edit sequence refactoring $\mathcal{D}'$ of $\mathcal{D}$ with size $sN$.

## 2.4 PROPERTIES OF LINTSEQ DATA

Synthetic edit sequences generated by LintSeq have a few other important properties. Let $\boldsymbol{\delta}_y$ be an arbitrary $j$-length edit sequence in $\mathcal{D}'$ generated with LintSeq, $\boldsymbol{\delta}_y = (\Delta(\varepsilon, y_1), \ldots, \Delta(y_{j-1}, y))$. First, we observe that there is a simple correspondence between $\boldsymbol{\delta}_y$ and the original program $y$ used to generate it: $y$ can be re-constructed by starting with an empty program, and successively applying each edit in $\boldsymbol{\delta}_y$ to this program one-by-one. In other words, the edit sequence $\boldsymbol{\delta}_y$ *resolves to* $y$. Furthermore, by construction, every prefix subsequence of $\boldsymbol{\delta}_y$ resolves to a intermediate program state of $y$ that is linter-error-free (see Section 2.1). These two properties, in conjunction with the uniform sampling step used in the first phase of the algorithm, show that LintSeq samples $s$ examples across all possible linter-error-free sequences of line insertions that can be used to sequentially write a program $y$ from-scratch.

We show an example of program synthesis dataset statistics before and after LintSeq processing in Appendix A (Figure 6). In the worst case, re-expressing a program as an edit sequence increases the length of a training example by a token count that is constant in the number of program lines[1].

## 2.5 PRACTICALITIES OF TRAINING LANGUAGE MODELS ON LINTSEQ DATA

LintSeq can be run on any code data. It is agnostic to the contents of a program, and only depends on knowledge of the language that a program is written in, and the existence of a linter for this language.

We use teacher-forced supervised learning (Williams & Zipser, 1989) to train models on LintSeq data, concatenating edit sequences into a single string by interleaving edits with special tokens, "`<|diff|>`," and computing instruction-conditioned losses over the resultant sequences. At test-time, fine-tuned models can be prompted to synthesize programs with edit sequences by appending these special tokens to the ends of prompts. More details are provided in Appendix B.

---

[1]See Appendix B for more details.

Synthetic data generation with LintSeq is controlled by a single hyperparameter: the number of edit sequences $s$ that are sampled for each example in the source code dataset $\mathcal{D}$. Edit sequence sampling can be constrained to avoid repetitions.

## 3 EXPERIMENTS

To study LintSeq and the impact of re-parameterizing program synthesis as a sequential edit generation problem, we conduct a set of supervised fine-tuning (SFT) experiments. These experiments study code synthesis in Python and are designed to answer the following questions:

- How does fine-tuning tiny code LMs to generate programs edit-by-edit with supervised learning impact performance on benchmarks compared to fine-tuning on standard code data?
- Do performance improvements hold for "off-the-shelf" LMs and on harder coding benchmarks? Do they hold across model scales, tokenizers, and families?
- How does ablating linter-guidance from LintSeq impact test-time performance?

Similar to previous works (Chen et al., 2021), we evaluate models by computing "pass@k," the probability that at least one of "k" generations for a problem passes all of the unit tests.

### 3.1 PRETRAINING TINY LMS FOR CODE UNDERSTANDING

We begin our investigations by pre-training two tiny decoder-only transformers, TinyCodeLM-150M and TinyCodeLM-400M, for Python code understanding on 72 billion tokens of text. Pretraining our own language models grants us a data contamination-free test-bed to study code synthesis with edit sequences, rapidly evaluate LintSeq, and broadly re-examine the trade-off between test-time compute and generation quality in code synthesis for models that can be updated on-device.

We rely on open-source data and libraries to pretrain our models (Penedo et al., 2024; Lozhkov et al., 2024; Soldaini et al., 2024; Groeneveld et al., 2024). Our pretraining data mix is inspired by Code Llama (Roziere et al., 2023), and reflects a code-skewed mixture of web text and raw Python sampled from FineWeb and The Stack, respectively (Penedo et al., 2024; Li et al., 2023). The architecture of our models respectively mimics the two smallest versions of GPT-2 (Radford et al., 2019), but integrates the transformer architecture changes proposed by the OLMo framework. This includes the absence of bias terms and the addition of non-parametric layer norms (Ba, 2016), as well as the use of SwiGLU (Shazeer, 2020), rotary positional embeddings (Su et al., 2024), and the GPT-NeoX-20B tokenizer (Black et al., 2022). We train both models for two epochs with a batch size of 524,288 tokens on an NVIDIA H100 node with four GPUs. Our experiments are supported by Pytorch FSDP (Zhao et al., 2023). More details on our pretraining procedures are in Appendix D.

### 3.2 GENERATING A SYNTHETIC DATASET WITH LINTSEQ

To support our fine-tuning experiments, we prepare a baseline dataset of paired instruction and program data. We then re-express the programs in this dataset as code edit sequences with LintSeq.

To that end, we first pool the Python portions of two open-source instruction datasets for code synthesis: the GPT 3.5/4-based Magicoder instruction dataset and the StarCoder2-15B-based self-alignment training dataset (Wei et al., 2024b;a). These datasets are generated with the OSS-Instruct approach by Wei et al. (2024b) and have undergone decontamination for the benchmarks that we evaluate on in this paper. We conduct de-duplication on the pooled data to check for repeated examples. Furthermore, we strip any chain-of-thought-like natural language explanations from completion data. The resultant dataset has over 88,900 instruction+program pairs.

With our baseline dataset prepared, we run LintSeq to generate $s = 5$ synthetic edit sequences for each instruction-program pair. As described in Section 2.5, we concatenate each synthetic edit sequence into a single string by interleaving consecutive edits with a special reserved "edit" token. Inspired by Muennighoff et al. (2024), we do not restrict against edit sequence repetitions. We use the popular Python linter `pylint` to guide edit sampling during generation. Examples of generated edit sequences and experiments testing the effect of varying $s$ are in Appendix F.

Table 1: **HumanEval and MBPP(+) results for TinyCodeLMs after SFT vs existing code models of similar scale** ($\leq$ 0.4B parameters). Scores annotated with "†" indicate external model evaluations that we ran using the procedure described in Appendix C, and all other scores are as reported by model authors. We list models in order of increasing HumanEval pass@1 and report standard error in computed score. Sampling hyperparameters are listed in Appendix C.4.

| Model | Size | HumanEval | | MBPP(+) | | Open-Source |
|---|---|---|---|---|---|---|
| | | pass@1 | pass@10 | pass@1 | pass@10 | |
| AlphaCode | 89M | 4.3 | 12.2 | - | - | ○ |
| Codex | 85M | 8.2 | 12.8 | - | - | ○ |
| SmolLM-Instruct | 135M | $7.7 \pm 0.8$† | $14.5 \pm 1.0$† | $10.1 \pm 1.8$† | $14.6 \pm 0.5$† | ● |
| TinyCodeLM-Instruct | 150M | $9.1 \pm 2.3$ | $13.5 \pm 0.6$ | $11.5 \pm 1.9$ | $21.6 \pm 0.4$ | ● |
| TinyCodeLM-Instruct | 400M | $11.3 \pm 0.9$ | $18.5 \pm 1.1$ | $15.5 \pm 2.1$ | $22.2 \pm 0.5$ | ● |
| SmolLM-Instruct | 360M | 11.3 | $19.3 \pm 1.1$† | $19.4 \pm 2.4$† | $23.1 \pm 0.5$† | ● |
| AlphaCode | 302M | 11.6 | 18.8 | - | - | ○ |
| CodeT5+ | 220M | 12.0 | 20.7 | - | - | ● |
| TinyCodeLM-LintSeqInstruct | 150M | **12.8** $\pm 2.6$ | $20.6 \pm 1.1$ | $13.6 \pm 2.1$ | $24.4 \pm 0.8$ | ● |
| Codegen-Mono | 350M | **12.8** | **23.1** | $9.4 \pm 1.8$† | $15.2 \pm 0.7$† | ● |
| Codex | 300M | **13.2** | 20.4 | - | - | ○ |
| TinyCodeLM-LintSeqInstruct | 400M | **13.4** $\pm 2.0$ | $20.9 \pm 1.1$ | **19.4** $\pm 2.4$ | **29.9** $\pm 0.6$ | ● |

### 3.3 TRAINING LANGUAGE MODELS ON LINTSEQ EDIT SEQUENCES WITH SFT

Next, we probe the impact of training autoregressive LMs to synthesize full programs vs. program edit sequences according to natural language instructions. Aside from the tiny code LMs described above in Section 3.3.1, we also finetune small LMs from three different model families, ranging in scale from 2.6B to 14B parameters. We evaluate tiny code LMs on HumanEval (Chen et al., 2021) and MBPP (Austin et al., 2021), and small LMs on the additional challenging benchmarks DS-1000 (Lai et al., 2023), BigCodeBench (Zhuo et al., 2024), and CodeContests (Li et al., 2022b).

Using both the refactored and baseline instruction datasets described in section 3.2, we run pairs of SFT experiments with six different models. In each experiment pair, we finetune an LM on both datasets for an equal number of optimizer steps and with the same learning rate schedule, saving intermediate checkpoints throughout fine-tuning. Then, we compare the benchmark performance of checkpoints across sampling temperatures[2], performing no prompt tuning. A more detailed description of the computed metrics as well as a full specification of the evaluation and fine-tuning procedures is provided in Appendices C and E.

### 3.3.1 TINYCODELM

We run our first two pairs of fine-tuning experiments on TinyCodeLM-150M and TinyCodeLM-400M. Our experimental results are summarized in Table 1, where we compare the temperature-tuned performance of our models on HumanEval and MBPP(+) to the pass@1 and pass@10 scores of existing LMs with similar parameter counts.

For both the 150M and 400M parameter versions of TinyCodeLM, we find that fine-tuning LMs to synthesize code with edits via LintSeq data results in stronger benchmark performance compared to the baseline, improving HumanEval pass@1 by 41% ($9.1 \mapsto 12.8$) and 19% ($11.3 \mapsto 13.4$) and MBPP pass@1 by 18% ($11.5 \mapsto 13.6$) and 25% ($15.5 \mapsto 19.4$). We see a similar scale of improvement on pass@10 for both benchmarks. Our smaller LintSeq model is particularly strong for its size, roughly matching the performance of several models with larger parameter counts (Table 1).

### 3.3.2 GEMMA 2, PHI-3, AND LLAMA 3.1

The results above raise a few questions: Do performance improvements from fine-tuning LMs to synthesize code with edit sequences also hold for language models that were not specifically pretrained for code understanding? Do they hold across model scales, architectures, and tokenizers?

---

[2]To process the generations of edit sequence LMs into executable programs, we simply resolve each of the predicted code edits one-by-one. This procedure is visualized in Figure 1 and described in Appendix B.2.

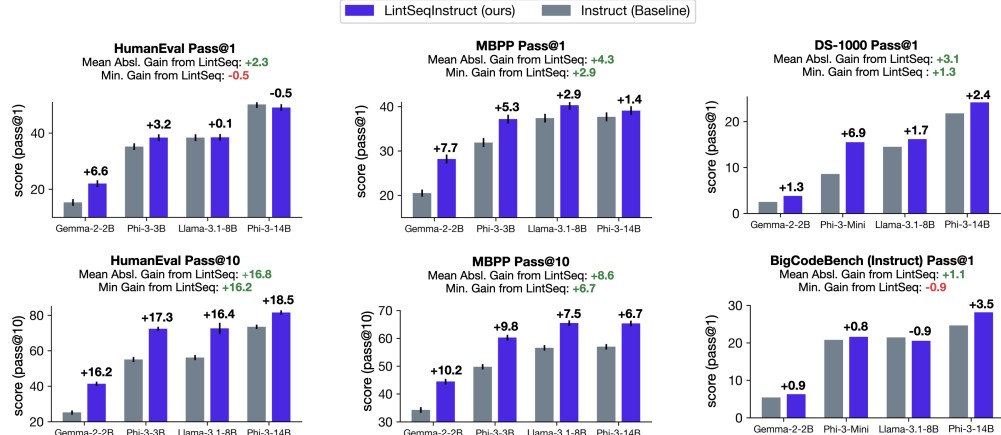

Figure 3: **HumanEval, MBPP(+), DS-1000, and BigCodeBench (Instruct) results for Gemma 2, Phi-3, and Llama 3.1 models after SFT on LintSeq (indigo) vs standard Python code (grey)**. On HumanEval and MBPP(+), we tune sampling temp., top-p, and min-p over $\{1, 1.1, 1.2\}$, $\{0.95, 1.0\}$, and $\{0, 0.05\}$, respectively with $n = 64$ samples. On DS-1000, we evaluate models with the completion format, temperature $= 0.2$, top-p $= 0.5$, min-p $= 0$, and $n = 40$, following Wei et al. (2024b) and Luo et al. (2023). On BigCodeBench Instruct, we evaluate with greedy decoding (Zhuo et al., 2024). Error bars on HumanEval and MBPP scores show standard error.

To answer these questions, we conduct four additional pairs of SFT experiments on LMs from three model families, Gemma 2, Phi-3, and Llama 3.1. We use pretrained-only model weights, if available. The selected LMs range in size from 2.6B to 14B and were trained on general-purpose data mixtures (Gemma Team et al., 2024; Abdin et al., 2024; Dubey et al., 2024).

Our findings align with those presented in Section 3.3.1. As shown in Figure 3, LintSeq improves performance on each LMs for all but two of the metrics visualized here (HumanEval pass@1 and BigCodeBench Instruct greedy pass@1). Notably, even on these metric, the least performant LintSeq instruction-tuned models still achieve performance that is comparable to the baseline, i.e. within standard error of sampling or within a percentage point. In aggregate across models, LintSeq improves HumanEval, MBPP, DS-1000, and BigCodeBench Instruct pass@1 by an average absolute gain of +2.3, +4.3, +3.1, and +1.1 in score compared to baseline SFT.

Furthermore, as shown in Figure 1(right) and Figure 4, the degree by which edit sequence LMs outperform baselines on HumanEval, MBPP, and CodeContests increases with repeated sampling for all tested models. In each of the plots included in these figures, we show the total proportion of benchmark problems solved by SFT-ed LMs on any attempt given "k" tries as a function of total test-time compute used during repeated sampling. By comparing total test-time compute across model variants, we account for the slight difference between LintSeqInstruct vs Instruct model generation lengths due to the extra "diff" descriptor tokens used by edit sequence models. Even after adjusting for these extra tokens, LintSeq consistently improves the relationship between total test-time compute and performance on code synthesis, supporting the hypothesis posed in Section 2.

In summary, the results of these experiments suggest that refactoring code tuning data into synthetic edit sequences with LintSeq is a code-pretraining-, scale-, architecture-, and tokenizer-independent mechanism for improving the quality and diversity of LM outputs on code generation tasks.

## 3.4 ABLATING THE LINTER FROM LINTSEQ

The backward sampling phase of LintSeq uses a linter to decompose code across edits whose contents reflect the syntactical structure of its programming language. We conclude our experiments by testing the importance of this design choice with TinyCodeLM models: does fine-tuning on sequences of (entirely) randomly sampled code edits hurt model performance on HumanEval and MBPP(+)?

To test this, we replace the backwards procedure described in Section 2.3 with fully random sampling; during each step of the algorithm, we first sample the number of lines to delete from the current

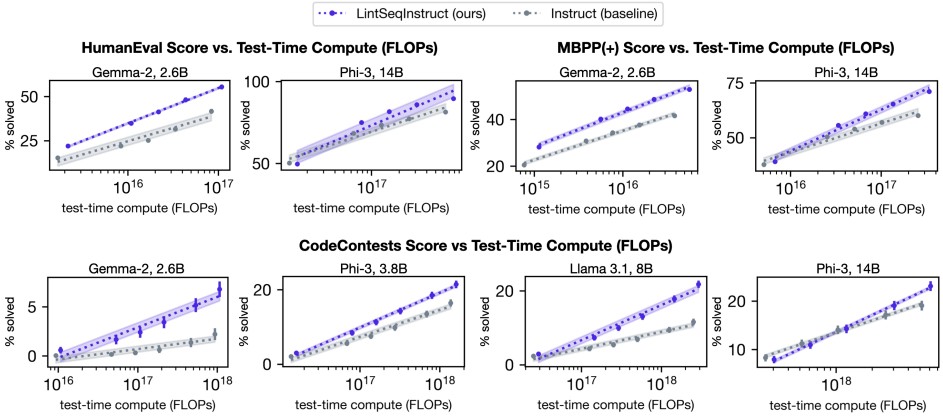

Figure 4: **Repeatedly sampling from models SFT-ed to generate edit seqs. vs full programs**: we compare the best pass@k score achieved by modulating sampling hyperparameters for LintSeqInstruct vs Instruct models. On HumanEval and MBPP(+), we use the same values as in Figure 3, while on CodeContests, we sweep over temperatures $\{0.5, 0.6\}$ and use top-p $= 1.0$, min-p $= 0$, and $n = 128$. We then plot benchmark score as a function of the total cost of repeated sampling from each model in FLOPs (see Appendix A.4). Shading shows standard error in linear fit. See Figure 1 for Phi-3 3.8B and Llama 3.1 8B test-time scaling with repeated sampling curves on HumanEval and MBPP.

program uniformly at random, before sampling a set of lines with the desired count. We refer to this algorithm as "RandSeq." Using RandSeq, we generate a new synthetic edit sequence dataset with the same size as the LintSeq dataset used in all previous fine-tuning experiments. The average number of edits per example in this dataset ($\approx 3.9$) is similar to its linter-guided counterpart ($\approx 3.8$)[3].

We employ the same procedure as the one used in Section 3.3 to SFT TinyCodeLM models on the RandSeq dataset. In Figure 5(left), we compare the pass@1 HumanEval and MBPP score of LintSeqInstruct vs RandSeqInstruct models at high temperatures. On both benchmarks and models, ablating the linter from LintSeq hurts performance with statistical significance, reducing HumanEval pass@1 by 30% ($6.4 \mapsto 4.5$) and 29% ($8.4 \mapsto 6.0$) and MBPP pass@1 by 24% ($8.6 \mapsto 6.5$) and 28% ($14.2 \mapsto 10.2$), respectively. These results suggest that the linter-informed structure of edits in LintSeq fine-tuning data does improve model performance.

In Figure 5(right), we conclude our analysis by probing whether training models on linted edits has an effect on the total proportion of syntactical errors in *completed* programs. To assess this, we run the Python linter `pylint` over the full set of generations sampled at temperature $= 1$, top-p $= 1$, and min-p $= 0$, checking each generated program for syntax errors with this linter. LMs trained on randomly sampled edits appear to generate "buggy" code with much higher frequency than all other models on both HumanEval and MBPP(+). Furthermore, on HumanEval, we find that LintSeq models synthesize programs with linter-errors at a higher frequency than baselines, despite their higher pass@1. This additional finding suggests that model performance gains from LintSeq cannot simply be attributed to improvement in low-level correctness of generated code – training on refactored code must be helping models write generally better, more diverse programs.

## 4 RELATED WORK

**Foundation Models for Code**  Code synthesis is one of the oldest problems in computer science. Neural language model-based approaches such as Codex, AlphaCode, CodeT5+, CodeGen, StarCoder, and Code Llama have recently proven to be extremely competitive with previous methods (Chen et al., 2021; Li et al., 2022b; Wang et al., 2023b; Nijkamp et al., 2022; Li et al., 2023; Roziere et al., 2023). Today, foundation models trained on web text and code data dominate, and LLM-powered code editing tools like Github Copilot and Cursor are used by thousands of engineers every day (Heaven, 2024). Many general-purpose LLMs are also trained on code data. While the largest of these LLMs show strong performance on coding benchmarks, generations continue to suffer from limited

---

[3]Note that both datasets also have a similar size in total training tokens ($\approx 18 \cdot 10^6$ TinyCodeLM tokens).

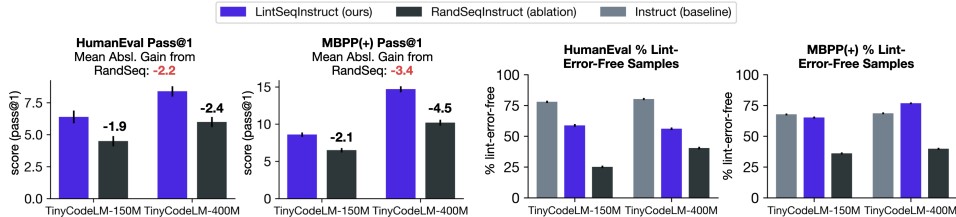

Figure 5: Left: HumanEval and MBPP(+) pass@1 achieved by fine-tuning TinyCodeLM models on **linter-guided (LintSeq) vs randomly sampled (RandSeq) code edit sequences**. We tune sampling parameters over the same values as in Figures 3 and 4, and report the best scores for each model. Right: Comparing **total proportions of generations with lint errors**. Error bars show standard error.

meaningful output diversity, prompt sensitivity, and degrading quality on long-contexts (Achiam et al., 2023; Gemini Team et al., 2023; Dubey et al., 2024). Smaller models also lag behind (Abdin et al., 2024; Gemma Team et al., 2024; Ben Allal et al., 2024). As of the writing of this paper, directly prompting LLMs to generate code "diffs" results in low quality edits across models (Sanger, 2024). We claim that this is the result of a data problem and we attempt to address it in this work.

**Finetuning on Synthetic Data**   LLM post-training methods like supervised finetuning have been shown to be extremely powerful for improving model performance across tasks (Wei et al., 2021). However, high-quality datasets of paired instruction-response examples are extremely expensive to curate. One possible solution lies in synthetic data generation methods like Self-Instruct, wherein an LLM is prompted to generate instructions and/or responses from examples (Wang et al., 2022). Such data have been used extensively for improving LLM performance through self-refinement and/or knowledge distillation on coding tasks (Chaudhary, 2023; Roziere et al., 2023; Abdin et al., 2024; Lozhkov et al., 2024). We employ post-processed instruction data for code synthesis created with a method from this family, OSS-Instruct (Wei et al., 2024b), as the base of our experiments on re-factorizing code with code edit sequences via LintSeq. Unlike Self-Instruct-like synthetic data generation methods, our algorithm does not employ an LLM for data generation, and instead generates examples of error-free edit sequences from existing code data by using a simple linter.

**Training on Edits**   Many works have studied edit generation with language models. Yin et al. (2018) cast the edit representation problem as an autoencoding task and show that neural network models can learn to capture the structure and semantics of edits, while Gu et al. (2019) introduce a partially autoregressive model for generating insertion and deletion edits that is trained with adversarial imitation learning. Guo et al. (2021) use reinforcement learning to train LMs to generate code with "holes" that represent high uncertainty tokens, and to edit the contents of these "holes" later on.

More recently, several works have investigated finetuning off-the-shelf pre-trained language models on large-scale edit data. Berabi et al. (2021) use a linter to detect errors in code, and finetune a T5 model (Raffel et al., 2020) to correct code by leveraging error messages. Muennighoff et al. (2023) and Cassano et al. (2023) instruction tune models on datasets of GitHub commits pairing code changes with human instructions. Relatedly, Li et al. (2024) use GitHub commit data sourced from Python repositories to generate code editing instruction data with GPT 3.5/ChatGPT. All of these works specifically focus on better-equipping LMs for natural language-prompted code editing tasks, in which a model is *explicitly* prompted to generate an edit in response to an error message or a natural language specification. Our work differs in three important ways: first, we study edit sequences rather than single edits; second, we train LMs to predict edits *implicitly* during code synthesis; third, our synthetic edit generation algorithm does not rely on the existence of any kind of commit data.

**"On Device" Language Models**   As the capabilities of LLMs have improved, so to have those of small language models. Recent projects like SmolLM (Ben Allal et al., 2024) and OpenELM (Mehta et al., 2024) re-examine the potential of tiny language models that can be run and even updated "on-device," i.e. on a smart phone or laptop. The representations learned by such models during pretraining are weaker than those of scaled-up LLMs (Kaplan et al., 2020). This is particularly true for harder tasks that involve reasoning, such as code synthesis (Gemma Team et al., 2024; Abdin et al., 2024). To our knowledge, the most recent open-source work studying small language models pretrained entirely for code understanding is from several years ago (Xu et al., 2022; Nijkamp et al., 2022; Wang et al., 2021; 2023b). The 150M and 400M parameter TinyCodeLM models

pretrained in this paper belong to the "on device" model family and build upon previous works. These models provide an efficient test-bed for experiments on LM code synthesis that is updated to recent advancements in high throughput pretraining and to improvements in open-source data quality.

**Scaling Up Test-Time Compute** The performance of language models can be boosted during inference by using scaled-up sample counts, hand-engineered prompting schema, and/or search (Brown et al., 2024; Snell et al., 2024). These methods dramatically increase inference costs. Their effectiveness is tightly linked to the expressivity of learned model representations and the diversity of outputs across samples. Our experiments with smaller language models are inspired by these works – we study whether it is possible to (1) improve the expressivity of representations for code synthesis across LM parameter scales during finetuning, and (2) take advantage of this property to improve the inference-time performance of smaller LMs by larger margins during repeated sampling.

## 5 DISCUSSION, LIMITATIONS, AND CONCLUSION

This paper introduces an algorithm, LintSeq, for generating synthetic code edit sequences from existing programs. LintSeq enables code synthesis to be re-parameterized at the data-level as sequential edit generation tasks. The algorithm is parameter-free, requires only CPU to run, and makes no assumptions about the content or structure of source code files.

Re-parameterizing code generation with edits has a few immediate benefits. For example, it makes code generation with LMs more controllable at the prompt-level (Appendix B.3) and it reduces the cost of predicting useful and syntactically correct code insertions with models, since synthetic edit-trained LMs do not need to be prompted to re-generate full programs from scratch (Section 2.5).

In our experiments with LintSeq, we also show the following:

1. Tiny LMs pre-trained for code understanding can be efficiently fine-tuned to synthesize programs edit-by-edit via LintSeq data. This results in competitive performance on HumanEval and MBPP(+) compared to existing code LMs of similar scale (Sections 3.1 and 3.3.1).

2. On larger models from the Phi 3, Gemma 2, and Llama 3.1 families that were pretrained for general natural language understanding, tuning on LintSeq data either improves or preserves the quality of pass@1 generations compared to standard tuning (Section 3.3.2).

3. LintSeq also improves test-time compute scaling laws for code synthesis on instruction fine-tuned Phi 3, Gemma 2, and Llama 3.1 models, suggesting that edit sequence LMs consistently generate more meaningfully diverse programs compared to baselines, even on challenging benchmarks like CodeContests (Section 3.3.2).

4. Ablating the linter from LintSeq hurts the quality and syntactical correctness of code synthesized by edit sequence TinyCodeLMs. This suggests that the structured nature of edits sampled with LintSeq is important for downstream LM performance (Section 3.4).

There are several limitations to our work.

First, as currently formulated, LintSeq can only be used to generate synthetic sequences of insertion edits. This is a consequence of the parameter-free nature of the algorithm – every edit in a LintSeq sequence reflects an existing line of code in the source file used to generate it. As a result, models that are fine-tuned exclusively on data sampled with LintSeq cannot be used for code editing tasks involving deletion edits. One simple way to circumvent this limitation might be by mixing LintSeq synthetic edit sequences with human edit data during instruction fine-tuning via datasets like CommitPackFT (Muennighoff et al., 2023), which contain examples of deletions. An alternate approach might be to follow-up supervised instruction fine-tuning on LintSeq synthetic data with reinforcement learning in order to train models to interleave insertions with deletions when necessary.

Second, the experiments that we conducted with LintSeq in this paper studied code synthesis in Python only. LintSeq can be similarly used for generating synthetic edit sequences for code written in other programming languages by swapping out the linter using during edit sampling.

Finally, we used LintSeq to refactor an instruction fine-tuning dataset in this work. However, by design, the algorithm can be run on *any* corpus of source code data, such as The Stack (Kocetkov et al., 2022) or The Stack-v2 (Li et al., 2023). In future work, we hope to explore using LintSeq to train LMs to write code edit-by-edit on larger, pre-training scale datasets.

## ETHICS STATEMENT

This work explores data-driven mechanisms for improving the quality of language model-generated code. Our synthetic data generation method relies on open-source data and our experiments leverage open-source software and resources. It is important to acknowledge that all language models for code synthesis have the potential to be misused – whether intentionally or unintentionally – for generation of code with vulnerabilities and/or malicious behaviors. Any and all model generated code has the potential to be harmful and must not be executed without precautions.

## REPRODUCIBILITY STATEMENT

In the supplementary materials accompanying this submission, we provide a Python implementation of LintSeq as well as instructions and code supporting data generation, processing, pretraining, and fine-tuning experiments. We also provide thorough textual descriptions of all experimental procedures in the Appendix. Appendix C describes prompting and model evaluation, while Appendices D and E detail all of the hyperparameters, procedures, and open-source datasets that we employ for obtaining the results reported throughout Section 3. Finally, Appendix A.4 provides references and data for reproducing the results plotted in Figure 1.

## ACKNOWLEDGEMENTS

This work was supported by grants from NSF award 2339096 and ONR awards N00014-21-1-2758 and N00014-22-1-2773. We are grateful to Shenglong Wang and NYU High Performance Computing for their support of this project. UP is funded by an NSF GRFP Award, and LP is funded by the Packard Fellowship. We would like to thank Nate Rahn, Mahi Shafiullah, and David Brandfonbrener for helpful comments and discussions.

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

# A    ADDITIONAL RESULTS

## A.1    EMPIRICS OF PROCESSING CODE DATA WITH LINTSEQ

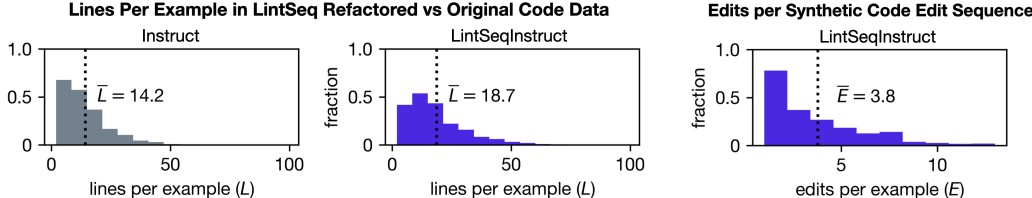

Figure 6: **Empirics of processing code data with LintSeq**. Left: Lines per example in a dataset of instruction fine-tuning data for Python synthesis before and after processing with LintSeq via the linter `pylint` (see Section 3.2). LintSeq processing adds lines of `diff` metadata to examples (see Appendix B). Right: The corresponding edit counts per synthetic code edit sequence. On a dataset of short programs (14 lines of code, on average), the mean LintSeq edit sequence contains four edits.

## A.2    COMPARING LINTSEQINSTRUCT TO RANDSEQINSTRUCT TINYCODELMS ON HUMANEVAL AND MBPP(+)

Table 2: **Edit sequence TinyCodeLM** results on **HumanEval at high sampling temperatures**: We tune sampling parameters for edit sequence variants of TinyCodeLM over temperatures (1, 1.1, 1.2), top-p (0.95, 1.0), and min-p (0, 0.05) with $n = 64$ completions per problem and report the best pass@k value obtained from each model variant. We also report standard error for each score.

| Model Variant | Size | Linter Guided | HumanEval | | | | |
|---|---|---|---|---|---|---|---|
| | | | pass@1 | pass@5 | pass@10 | pass@20 | pass@50 |
| tinycodeLM-RandSeqInstruct | 150M | ✗ | $4.5 \pm 0.4$ | $10.3 \pm 0.5$ | $12.2 \pm 0.5$ | $14.4 \pm 0.6$ | $18.8 \pm 0.6$ |
| tinycodeLM-LintSeqInstruct | 150M | ✓ | $\mathbf{6.4} \pm 0.5$ | $\mathbf{13.9} \pm 0.5$ | $\mathbf{16.8} \pm 0.6$ | $\mathbf{19.5} \pm 0.6$ | $\mathbf{23.6} \pm 0.6$ |
| tinycodeLM-RandSeqInstruct | 400M | ✗ | $6.0 \pm 0.4$ | $11.7 \pm 0.5$ | $13.9 \pm 0.6$ | $16.4 \pm 0.6$ | $20.8 \pm 0.6$ |
| tinycodeLM-LintSeqInstruct | 400M | ✓ | $\mathbf{8.4} \pm 0.4$ | $\mathbf{16.6} \pm 0.6$ | $\mathbf{19.7} \pm 0.6$ | $\mathbf{22.8} \pm 0.6$ | $\mathbf{27.2} \pm 0.6$ |

Table 3: **Edit sequence TinyCodeLM** results on **MBPP(+) at high sampling temperatures**: As above, we tune sampling parameters for all fine-tuned TinyCodeLM variants over temperatures (1, 1.1, 1.2), top-p (0.95, 1.0), and min-p (0, 0.05) with $n = 64$ completions per problem and report the best pass@k value obtained from each model variant. Standard error is indicated with "$\pm$."

| Model Variant | Size | Linter Guided | MBPP(+) | | | | |
|---|---|---|---|---|---|---|---|
| | | | pass@1 | pass@5 | pass@10 | pass@20 | pass@50 |
| tinycodeLM-RandSeqInstruct | 150M | ✗ | $6.5 \pm 0.3$ | $17.2 \pm 0.4$ | $22.6 \pm 0.4$ | $27.9 \pm 0.5$ | $34.4 \pm 0.5$ |
| tinycodeLM-LintSeqInstruct | 150M | ✓ | $\mathbf{8.6} \pm 0.3$ | $\mathbf{19.5} \pm 0.4$ | $\mathbf{24.5} \pm 0.5$ | $\mathbf{29.0} \pm 0.5$ | $\mathbf{35.1} \pm 0.5$ |
| tinycodeLM-RandSeqInstruct | 400M | ✗ | $10.2 \pm 0.4$ | $20.8 \pm 0.4$ | $25.4 \pm 0.5$ | $29.9 \pm 0.5$ | $36.2 \pm 0.5$ |
| tinycodeLM-LintSeqInstruct | 400M | ✓ | $\mathbf{14.7} \pm 0.4$ | $\mathbf{25.8} \pm 0.5$ | $\mathbf{29.6} \pm 0.5$ | $\mathbf{33.9} \pm 0.5$ | $\mathbf{39.7} \pm 0.5$ |

### A.3 HUMANEVAL, MBPP(+), CODECONTESTS, DS-1000, AND BIGCODEBENCH RESULTS FOR LINTSEQ VS BASELINE INSTRUCTION TUNED GEMMA 2, PHI-3, AND LLAMA 3.1 MODELS

Table 4: **Gemma 2, Phi-3, and Llama 3.1** results on **HumanEval** at high sampling temperatures. We report the best pass@k value obtained from each model variant at high sampling temperatures, sweeping over temperature values (1, 1.1, 1.2), top-p (0.95, 1.0), and min-p (0, 0.05). We generate $n = 64$ completions per problem and report standard error for each estimated score.

| Model Variant | Size | HumanEval | | | | |
| --- | --- | --- | --- | --- | --- | --- |
| | | pass@1 | pass@5 | pass@10 | pass@20 | pass@50 |
| Gemma-2-Instruct | 2.6B | $15.3 \pm 0.6$ | $22.0 \pm 0.6$ | $25.2 \pm 0.6$ | $31.6 \pm 0.6$ | $41.7 \pm 0.7$ |
| Gemma-2-LintSeqInstruct | 2.6B | $\mathbf{22.0} \pm 0.6$ | $\mathbf{34.8} \pm 0.6$ | $\mathbf{41.4} \pm 0.6$ | $\mathbf{48.2} \pm 0.7$ | $\mathbf{55.5} \pm 0.7$ |
| Phi-3-Mini-Instruct | 3.8B | $35.2 \pm 0.6$ | $49.7 \pm 0.6$ | $55.1 \pm 0.7$ | $59.2 \pm 0.7$ | $62.2 \pm 0.7$ |
| Phi-3-Mini-LintSeqInstruct | 3.8B | $\mathbf{38.4} \pm 0.6$ | $\mathbf{63.3} \pm 0.6$ | $\mathbf{72.4} \pm 0.6$ | $\mathbf{79.9} \pm 0.6$ | $\mathbf{87.3} \pm 0.5$ |
| Llama-3.1-Instruct | 8B | $\mathbf{38.4} \pm 0.6$ | $51.3 \pm 0.7$ | $56.2 \pm 0.7$ | $60.2 \pm 0.7$ | $64.2 \pm 0.7$ |
| Llama-3.1-LintSeqInstruct | 8B | $\mathbf{38.5} \pm 0.6$ | $\mathbf{62.2} \pm 1.6$ | $\mathbf{72.6} \pm 1.6$ | $\mathbf{75.7} \pm 0.6$ | $\mathbf{82.7} \pm 0.6$ |
| Phi-3-Med-Instruct | 14B | $\mathbf{50.2} \pm 0.6$ | $68.4 \pm 0.6$ | $73.5 \pm 0.6$ | $77.3 \pm 0.6$ | $81.4 \pm 0.6$ |
| Phi-3-Med-LintSeqInstruct | 14B | $\mathbf{49.7} \pm 0.6$ | $\mathbf{75.0} \pm 0.6$ | $\mathbf{81.6} \pm 0.6$ | $\mathbf{85.9} \pm 0.6$ | $\mathbf{89.6} \pm 0.5$ |

Table 5: **Gemma 2, Phi-3, and Llama 3.1** results on **MBPP(+)** at high sampling temperatures. Exactly as above, we sweep over temperature (1, 1.1, 1.2), top-p (0.95, 1.0), and min-p (0, 0.05) and report the best pass@k value obtained from each model variant. We generate $n = 64$ completions per problem and report standard error for each estimated score.

| Model Variant | Size | MBPP(+) | | | | |
| --- | --- | --- | --- | --- | --- | --- |
| | | pass@1 | pass@5 | pass@10 | pass@20 | pass@50 |
| Gemma-2-Instruct | 2.6B | $20.5 \pm 0.4$ | $30.8 \pm 0.5$ | $34.3 \pm 0.5$ | $37.6 \pm 0.5$ | $41.6 \pm 0.5$ |
| Gemma-2-LintSeqInstruct | 2.6B | $\mathbf{28.2} \pm 0.5$ | $\mathbf{40.1} \pm 0.5$ | $\mathbf{44.5} \pm 0.5$ | $\mathbf{48.6} \pm 0.5$ | $\mathbf{52.8} \pm 0.5$ |
| Phi-3-Mini-Instruct | 3.8B | $31.9 \pm 0.5$ | $42.5 \pm 0.5$ | $46.3 \pm 0.5$ | $49.8 \pm 0.5$ | $53.6 \pm 0.5$ |
| Phi-3-Mini-LintSeqInstruct | 3.8B | $\mathbf{37.2} \pm 0.5$ | $\mathbf{51.4} \pm 0.5$ | $\mathbf{56.1} \pm 0.5$ | $\mathbf{60.3} \pm 0.5$ | $\mathbf{66.0} \pm 0.5$ |
| Llama-3.1-Instruct | 8B | $37.4 \pm 0.5$ | $50.2 \pm 0.5$ | $53.6 \pm 0.5$ | $56.6 \pm 0.5$ | $60.0 \pm 0.5$ |
| Llama-3.1-LintSeqInstruct | 8B | $\mathbf{40.3} \pm 0.5$ | $\mathbf{56.2} \pm 0.5$ | $\mathbf{61.1} \pm 0.5$ | $\mathbf{65.5} \pm 0.5$ | $\mathbf{69.4} \pm 0.5$ |
| Phi-3-Med-Instruct | 14B | $37.7 \pm 0.5$ | $50.4 \pm 0.5$ | $54.0 \pm 0.5$ | $57.0 \pm 0.5$ | $60.1 \pm 0.5$ |
| Phi-3-Med-LintSeqInstruct | 14B | $\mathbf{39.1} \pm 0.5$ | $\mathbf{55.2} \pm 0.5$ | $\mathbf{60.7} \pm 0.5$ | $\mathbf{65.4} \pm 0.5$ | $\mathbf{71.1} \pm 0.5$ |

Table 6: **Gemma 2, Phi-3, and Llama 3.1** results on **CodeContests**. We sweep over temperature (0.5, 0.6) and use top-p = 1, min-p = 0, and $n = 128$, and report the best pass@k value obtained from each model variant in the table below. We also report standard error for each estimated score.

| Model Variant | Size | CodeContests | | |
| --- | --- | --- | --- | --- |
| | | pass@1 | pass@50 | pass@100 |
| Gemma-2-Instruct | 2.6B | $0.05 \pm 0.05$ | $1.56 \pm 0.26$ | $2.26 \pm 0.30$ |
| Gemma-2-LintSeqInstruct | 2.6B | $\mathbf{0.61} \pm 0.16$ | $\mathbf{5.71} \pm 0.37$ | $\mathbf{7.03} \pm 0.40$ |
| Phi-3-Mini-Instruct | 3.8B | $1.80 \pm 0.22$ | $14.86 \pm 0.45$ | $18.59 \pm 0.49$ |
| Phi-3-Mini-LintSeqInstruct | 3.8B | $\mathbf{2.76} \pm 0.26$ | $\mathbf{19.10} \pm 0.48$ | $\mathbf{22.93} \pm 0.51$ |
| Llama-3.1-Instruct | 8B | $\mathbf{2.68} \pm 0.28$ | $11.21 \pm 0.44$ | $12.80 \pm 0.46$ |
| Llama-3.1-LintSeqInstruct | 8B | $\mathbf{2.92} \pm 0.27$ | $\mathbf{17.86} \pm 0.47$ | $\mathbf{21.82} \pm 0.51$ |
| Phi-3-Med-Instruct | 14B | $\mathbf{3.22} \pm 0.27$ | $16.50 \pm 0.47$ | $19.45 \pm 0.50$ |
| Phi-3-Med-LintSeqInstruct | 14B | $\mathbf{3.02} \pm 0.25$ | $\mathbf{19.09} \pm 0.48$ | $\mathbf{23.11} \pm 0.51$ |

Table 7: **Gemma 2, Phi-3, and Llama 3.1** pass@1 results on **DS-1000**. We use the same sampling hyperparameters as Luo et al. (2023) and Wei et al. (2024b) to evaluate instruction tuned models.

| Model Variant | Size | DS-1000, pass@1 |
|---|---|---|
| Gemma-2-Instruct | 2.6B | 2.5 |
| Gemma-2-LintSeqInstruct | 2.6B | **3.8** |
| Phi-3-Mini-Instruct | 3.8B | 8.6 |
| Phi-3-Mini-LintSeqInstruct | 3.8B | **15.5** |
| Llama-3.1-Instruct | 8B | 14.5 |
| Llama-3.1-LintSeqInstruct | 8B | **16.2** |
| Phi-3-Med-Instruct | 14B | 21.8 |
| Phi-3-Med-LintSeqInstruct | 14B | **24.2** |

Table 8: **Gemma 2, Phi-3, and Llama 3.1** pass@1 results on **BigCodeBench (Instruct)**. We use greedy decoding to evaluate instruction tuned models.

| Model Variant | Size | BigCodeBench Instruct, pass@1 |
|---|---|---|
| Gemma-2-Instruct | 2.6B | 5.44 |
| Gemma-2-LintSeqInstruct | 2.6B | **6.32** |
| Phi-3-Mini-Instruct | 3.8B | 20.79 |
| Phi-3-Mini-LintSeqInstruct | 3.8B | **21.58** |
| Llama-3.1-Instruct | 8B | **21.46** |
| Llama-3.1-LintSeqInstruct | 8B | 20.53 |
| Phi-3-Med-Instruct | 14B | 24.65 |
| Phi-3-Med-LintSeqInstruct | 14B | **28.16** |

### A.4 COMPUTING PASS@K VS TOTAL TEST-TIME FLOPS

In Figures 1(right) and 4, we plot the percentage of problems solved by any attempt (i.e. pass@k) on HumanEval, MBPP, and CodeContests as a function of total test-time FLOPs used during sampling for LintSeq vs baseline instruction fine-tuned models. Raw "pass@k" estimates are also included in Tables 4, 5, and 8, representing the best scores achieved by each model variant after tuning sampling hyperparameters.

We compute total test-time FLOPs using the approximations below, which are drawn from Kaplan et al. (2020). These approximations conservatively estimate the cumulative inference costs of synthesizing solutions to all of the problems in the test set of each benchmark. The models that we compare are all dense transformers, where the majority of the parameters are used in matrix multiplications.

$$\text{FLOPs per token} \approx 2 \cdot (N_{\text{model-params}} + 2 \cdot L_{\text{model-layers}} \cdot C_{\text{context}})$$
$$\text{Total FLOPs} \approx \text{FLOPs per token} \cdot T_{\text{avg-total-tokens-per-sample}} \cdot K_{\text{samples}} \cdot M_{\text{problems}}$$

We determine the quantities $T_{\text{avg-total-tokens-per-sample}}$ for each model variant at a particular "pass@k" by computing token counts over all sets of samples per problem.

Note that edit sequence (i.e. LintSeqInstruct fine-tuned) LMs have slightly higher average token counts per sample due to presence of "diff" descriptor tokens in generations (see Appendix B).

# B  MORE ON EDIT SEQUENCES AND DIFFS

## B.1  READING UNIX DIFFS

We provide a guide to reading Unix-style diffs below in Figure 7. The diff shown in this figure is computed using the Python library `difflib`, which is the implementation that we use to compactly represent edits in our synthetic data generation experiments. Note that the total extra tokens present in an insertion edit sequence representation of a program scales with the number of program lines $L$, and can be upper-bounded as $T_{\text{diff}} \leq L \cdot ((\text{chars in "decorator"}) + (\text{extra chars per line in "body"}))$.

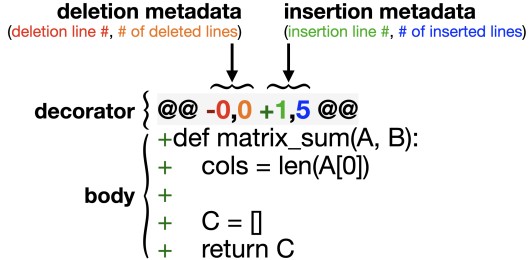

Figure 7: **The anatomy of a Unix diff**: A diagrammatic visualization of the different parts of a Unix-style diff, as computed by `difflib`. The *body* of a diff can consist of multiple line deletions, followed by multiple line insertions. The *decorator* portion of the diff shows the location and size of these deletions and insertions, if any. Like the diff shown above, the edits in synthetic edit sequences generated by LintSeq consist of line insertions only.

## B.2  RESOLVING EDIT SEQUENCES

During inference, LMs that have been fine-tuned on LintSeq instruct data will iteratively synthesize programs by generating edits i.e., outputting text that consists of a sequence of consecutive Python diffs interleaved with newline characters and "`<|diff|>`" tokens, similar to  Piterbarg et al. (2024). If correctly formatted by the LM, these diffs will be structured as shown in Figure 7.

Resolving an edit sequence generated by a language model into an executable Python program is simple: starting with an empty program, we consecutively apply the line insertions and/or deletions in the body of each diff to the lines of the program specified in its decorator. We continue this process until all of the diffs in the generated edit sequence have been parsed and resolved.

Figure 1 shows a code edit sequence generation from a LintSeq instruction fine-tuned LM and the corresponding resolved, executable Python program.

## B.3  CONTROLLABILITY OF CODE SYNTHESIS WITH EDIT SEQUENCE LMS

The structure of Unix-style diffs affects the downstream controllability of code synthesis with models that have been trained on edit sequence re-parameterized programs. As shown in Figure 7, the first line of every diff is a decorator that describes the location and the number of lines changed by the edit. During inference, autoregressive language models that have been trained on diffs with this format can be prompted to predict an edit in a target location by intervening on a model generation.

## B.4  FUTURE WORK: SEARCHING IN EDIT SPACE

If we apply the lens of reinforcement learning or search to this setting, we might say that re-parameterizing the code data used to train a language model re-parameterizes the model's action space. It is possible that combining edit sequence LMs with more sophisticated decoding mechanisms, test-time search, and/or reinforcement learning may result in even larger improvements to the quality of generated code than those of the zero-shot code synthesis settings studied in this paper. We look forward to testing this in future work.

## C  EVALUATION

HumanEval (Chen et al., 2021) and Mostly-Basic Programming Problems (MBPP) (Austin et al., 2021) are two of the most studied benchmarks for evaluating code LMs (Liu et al., 2023). These benchmarks probe the code synthesis capabilities of models, and consist of pairs of natural language program descriptions and test-cases. We employ the extended MBPP test cases released as MBPP(+) by Liu et al. (2023) to add additional rigour to our testing procedure. The code LMs that we compare our TinyCodeLM models against in Table 1 evaluate HumanEval performance using the original set of benchmark test cases; for consistency, we employ these same test cases in all of our evaluations.

Our evaluations on the harder benchmarks CodeContests, DS-1000, and BigCodeBench(Instruct) use exactly the same sets of problem descriptions and test cases as those introduced by Li et al. (2022b), Lai et al. (2023), and Zhuo et al. (2024).

During testing on each benchmarks, LMs are prompted to generate outputs using the natural language descriptions of target programs. Their outputs are then evaluated on the paired test cases. A generation is considered "correct" if and only if it passes all of the test cases upon execution, subject to a fixed timeout setting. Previous works on code synthesis with language models report scores across samples. The most common of these metrics is known as pass@k (Chen et al., 2021; Austin et al., 2021; Li et al., 2022b; Wang et al., 2023b). This is the metric that we use to report and compare model performance throughout this paper.

### C.1  PROMPTING

The primary goal of this paper is to introduce a method for re-factorizing code synthesis with LMs by fine-tuning them on synthetic instruction data. As a result, we evaluate all models using minimal prompt formats, performing no prompt tuning (see Figures 9 and 10). Examples of the prompt formats that we use during evaluation are shown in Figure 8.

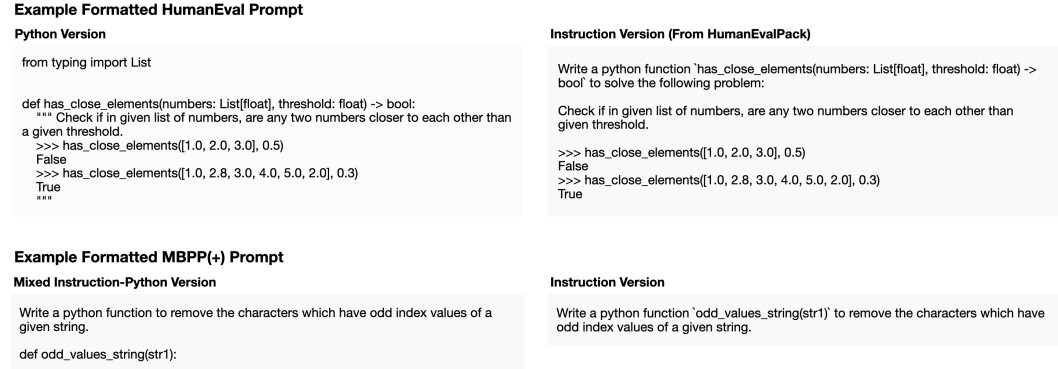

Figure 8: Examples of formatted HumanEval and MBPP(+) prompts used in model evaluations.

We finetune all tested models on example outputs exclusively corresponding to Python code, and as a result, we do not use Markdown formatting to separate Python code from natural language in either our instruction data nor in our inference-time prompts.

To evaluate models on HumanEval, we use both the default "Python version" prompt format in the original benchmark dataset, where a natural language program description is provided to an LM within a docstring, as well as the equivalent, fully natural language prompt format from HumanEvalPack (Muennighoff et al., 2023). The latter format is similar to the structure of the instructions in our fine-tuning datasets. We report results on the prompt format that yields the best score for each model.

To evaluate models on MBPP(+), we use the default prompts from the MBPP benchmark dataset, formatted with specification of the target function name and arguments both inside and outside of the natural language instruction, as shown in Figure 8. As on HumanEval, we report results on the prompt format that yields the best score for each model.

To evaluate models on BigCodeBench(Instruct) and CodeContests, we simply prompt models with the problem descriptions introduced in the original version of the benchmark (Zhuo et al., 2024; Li et al., 2022b).

Finally, to evaluate models on DS-1000, we use the completion format, with precisely the same prompt structures as those used by Wei et al. (2024b).

## C.2 GENERATION AND PARSING

During generation, we continue decoding until an end-of-sequence token is output by an LM. We treat all LM outputs as either Python code or sequences of Python code edits, depending on whether an LM was fine-tuned on standard instruct or LintSeq instruct data. In the latter case, we post-process outputs by resolving the output edit sequences using the procedure described in Appendix B.2.

## C.3 EVALUATING MODEL CHECKPOINTS

### C.3.1 PHILOSOPHY

There is a well-known trade-off between the temperature used for sampling from autoregressive code LMs and the benchmark coverage achievable by models, i.e. the proportion of problems "pass@k" for which an LM is able to generate at least one output that passes all test cases given "k" tries. This trade-off was first described by Chen et al. (2021). Informally, increasing the sampling temperature increases the width of the distribution from which tokens are sampled, producing more diverse but noisier (and possibly lower quality) generations. For larger repeated sample counts, the pass@k score typically increases with sampling temperature up to some threshold, beyond which the negative effects of noise overpower the positive effects of diversity. The benchmark coverage achievable by an LM at any temperature and in the limit of samples, i.e. on pass@k for $k \uparrow \infty$, ultimately depends on both the power and expressivity of the code language model's learned representation.

From a practical perspective, while smaller language models may have weaker representational power than larger models, the representational expressivity of the former may enable them to overtake the latter at fixed computational budgets by leveraging extra compute at inference-time, e.g. generating a larger number of samples per problem and using the provided test cases to check each one for correctness before returning an output (Brown et al., 2024; Snell et al., 2024). For example, an LLM that has an 85% pass@1 score on an arbitrary task may be more expensive in total serving cost (see Figure 1) than a smaller LM with a 90% pass@50 score on the same task. A small LM can only have this property, however, if it exhibits a reliable trade-off between generation quality and inference-time sampling cost across tasks. In other words, its representation must be sufficiently expressive.

### C.3.2 COMPUTING PASS@K

Our goal is to probe whether re-parameterizing code synthesis with edit sequences can improve the expressivity of smaller LM representations, boosting benchmark scores as a function of total test-time compute. Hence, we primarily compare fine-tuned models by evaluating them with the procedures described above across multiple pass@k. We compute unbiased pass@k statistics with the same procedure as Chen et al. (2021). The results of these evaluations are reported throughout the paper.

## C.4 COMPARING TINYCODELMS TO EXISTING MODELS IN TABLE 1

Many existing state-of-the-art code synthesis LMs only report temperature-tuned pass@k scores on HumanEval, including Codex, AlphaCode, and Codegen-Mono (Chen et al., 2021; Li et al., 2022b; Nijkamp et al., 2022). Thus, in Table 1, we temperature-tune TinyCodeLM models' pass@1 and pass@10 scores when reporting results. On HumanEval, we test temperatures $\tau \in \{0.0, 0.2, 0.4, 0.8, 1.0\}$. On MBPP(+), we sweep over a smaller temperature range, $\tau \in \{0.0, 0.1, 1.0\}$. We perform the same temperature tuning procedure when reporting external model benchmark scores as well, i.e. the scores annotated with "($\dagger$)" in Table 1. When running benchmark evaluations with these external code LMs, we stray from the prompt formatting, generation, and parsing procedures described in Appendices C.1 and C.2; instead, in the interest of a fair evaluation, we reproduce the conventions reported by model authors to report other scores.

# D PRETRAINING

We rely on data and libraries open-sourced by the HuggingFace, FineWeb, StarCoder, Dolma, OLMo, and PyTorch FSDP projects to pretrain our models (Wolf et al., 2020; Penedo et al., 2024; Lozhkov et al., 2024; Soldaini et al., 2024; Groeneveld et al., 2024; Zhao et al., 2023).

## D.1 MODEL ARCHITECTURES AND PRETRAINING HYPERPARAMETERS

Table 9: **Architectural and pretraining hyperparameters** of our "on device" 150M and 400M parameter TinyCodeLM models, pretrained on a mixture of Web text and code for Python understanding.

| | TinyCodeLM | |
|---|---|---|
| | Smallest, 150M Parameters | Small, 400M Parameters |
| Transformer Architecture | decoder-only | decoder-only |
| Model Family | `OlmoForCausalLM` | `OlmoForCausalLM` |
| Tokenizer | GPT-NeoX-20B-OLMo | GPT-NeoX-20B-OLMo |
| Attention Bias | False | False |
| Attention Dropout | 0.0 | 0.0 |
| Hidden Activation | `SwiGLU` | `SwiGLU` |
| Hidden Size | 768 | 1024 |
| Intermediate Size | 3072 | 4096 |
| Number of Attention Heads | 12 | 16 |
| Number of Hidden Layers | 12 | 24 |
| Number of Key-Value Heads | 12 | 16 |
| Vocabulary Size | 50304 | 50304 |
| Positional Encodings | Rotary (`RoPE`) | Rotary (`RoPE`) |
| Mixed Precision | `BFLOAT16` | `BFLOAT16` |
| Weight Tying | True | True |
| Flash Attention 2 | True | True |
| Optimizer | AdamW | AdamW |
| Learning Rate | 0.0003 | 0.0003 |
| Weight Decay | 0.01 | 0.01 |
| Betas | $(0.9, 0.95)$ | $(0.9, 0.95)$ |
| Epsilon | 1.0e-05 | 1.0e-05 |
| Learning Rate Scheduler | cosine (with warmup) | cosine (with warmup) |
| Number of Warm-Up Steps | 100 | 100 |
| Alpha-f ($\alpha_f$) | 0.1 | 0.1 |
| Total Epochs of Pretraining | 2 | 2 |

## D.2 PRETRAINING DATA MIX

Table 10: **Pretraining data mix** used to train both TinyCodeLM models. Datasets were tokenized and prepared using HuggingFace and Dolma tooling (Wolf et al., 2020; Soldaini et al., 2024).

| Pretraining Data Source | Subset | Tokens | Documents |
|---|---|---|---|
| FineWeb (Penedo et al., 2024) | 10BT Sample | 10.4BT | 14.9M |
| The Stack (Kocetkov et al., 2022) | Python Only | 61.8BT | 24.2M |

# E  INSTRUCTION FINE-TUNING

## E.1  BASELINE INSTRUCTION DATASET

Table 11 displays the data sources that are used to prepare the dataset described in Section 3.2. These data are pooled and preprocessed into instruction-program pairs by stripping away Markdown formatting and natural language explanations from completions (Figure 9 and 10). In our experiments, we use the resultant data to finetune baseline models, comparing their performance to those of LMs fine-tuned on edit sequences generated with LintSeq from the same set of instruction-program pairs.

| HuggingFace Instruction Data Source | Subset | Examples |
|---|---|---|
| bigcode/self-oss-instruct-sc2-exec-filter-50k | Full | 50,661 |
| ise-uiuc/Magicoder-OSS-Instruct-75K | Python | 38,284 |

Table 11: **Instruction data mix** used to prepare the baseline instruction dataset in Section 3.2.

## E.2  PROCEDURES AND HYPERPARAMETERS

We instruction finetune all models with Microsoft DeepSpeed using the ZeRO++ protocol for stage three sharding. For the largest of these models, we also use CPU parameter offloading to accelerate experiments (Wang et al., 2023a; Ren et al., 2021). When fine-tuning models on LintSeq data, we add a new token "<|diff|>" to tokenizers (Section 2.5) and resize model embeddings accordingly.

In our experiments with Gemma 2, Phi-3, and Llama 3.1 models, we use HuggingFace to access and load pretrained model weights and tokenizers. As mentioned in the main body of the paper, we instruction finetune *pretrained-only* weights if open-sourced and available. This is the case for Gemma 2 and Llama 3.1 only, as of the writing of this paper.

Across all of the fine-tuning experiments conducted in this paper, we train model-data variants with the same batch size and for an equal number of total optimizer steps. This optimizer step count corresponds to ten epochs of fine-tuning with the baseline instruction tuning dataset described in Section 3.2. We save intermediate checkpoints at equal optimizer step intervals in all experiments, and we report benchmark scores for the best performing checkpoint from each model-data variant.

In order to tune the peak learning rates used in each set of model experiments, we run a full sweep $\alpha \in \{$6e-4, 3e-4, 1e-4, 5e-5, 1e-5, 5e-6$\}$ in the baseline instruction data setting for each model. We select peak learning rate values by tracking the best-achieved downstream benchmark performance across models. The chosen values are displayed in Table 12. All other fine-tuning hyperparameters are kept fixed at the settings in Table 13 across experiments.

| | TinyCodeLM | | Gemma 2 | Phi-3 | | Llama 3.1 |
|---|---|---|---|---|---|---|
| | 150M | 400M | 2B | 3.8B | 14B | 8B |
| Peak Learning Rate ($\alpha$) | 3e-4 | 3e-4 | 5e-5 | 5e-5 | 1e-5 | 1e-5 |

Table 12: **Peak learning rates** used to instruction finetune models.

| | Hyperparameter Setting |
|---|---|
| Learning Rate Scheduler | linear |
| Max Learning Rate | 1e-4 |
| Warmup Ratio | 0.001 |
| Weight Decay | 0.01 |
| Total Batch Size | 512 |
| Batch Loss Reduction | sum |
| Mixed Precision | BFLOAT16 |
| Max Sequence Length | 1024 |
| Total Optimizer Steps | 1740 |

Table 13: **All other instruction fine-tuning settings**, re-used across experiments.

# F   MORE ON SYNTHETIC DATA GENERATION WITH LINTSEQ

## F.1   EXAMPLES OF GENERATED SYNTHETIC EDIT TRAJECTORIES

**Example A**

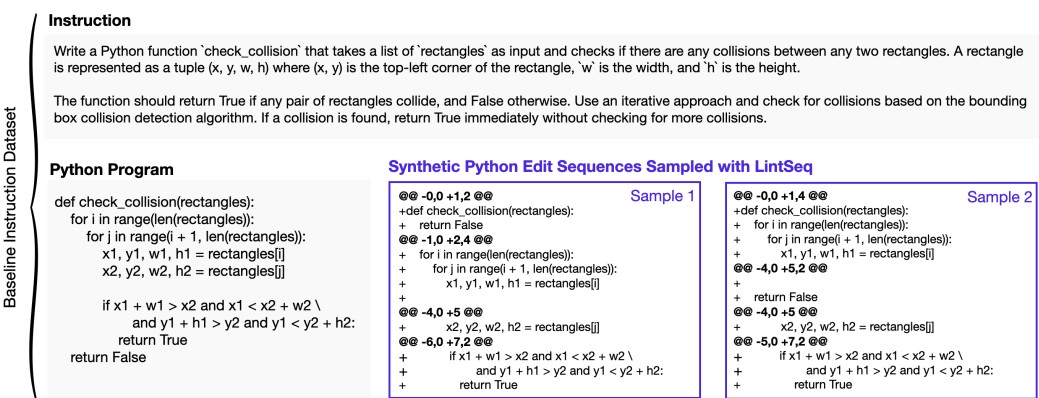

Figure 9: LintSeq edit sequence samples vs baseline instruction-program data, **example A**.

**Example B**

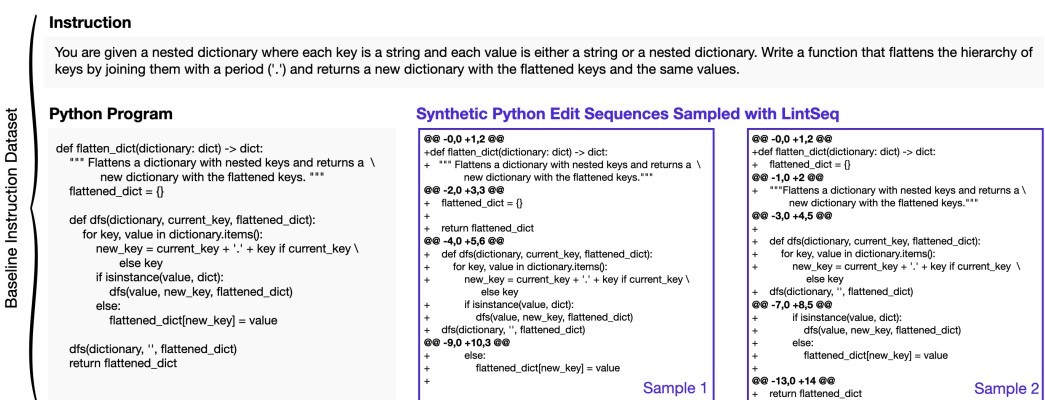

Figure 10: LintSeq edit sequence samples vs baseline instruction-program data, **example B**.

## F.2   TUNING LINTSEQ EXAMPLE COUNT

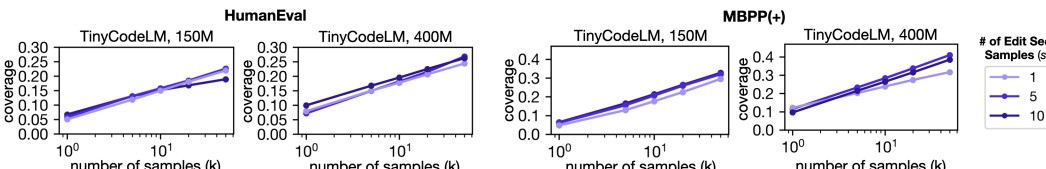

Figure 11: **Probing the effect of varying the number of edit sequences sampled with LintSeq per instruction-example pair during data generation**: Using the source dataset described in Section 3.2, we sweep over the value of the LintSeq parameter $s$ used during synthetic data generation to yield three different edit sequence instruction datasets with $s \in \{1, 5, 10\}$. We finetune TinyCodeLM models on each of these datasets, and compare the resultant HumanEval and MBPP(+) performance vs samples (i.e. pass@k vs k) at temperature 1. The most performant values is $s = 5$.

