# OpenReview forum: "Training Language Models on Synthetic Edit Sequences Improves Code Synthesis"
_ICLR.cc/2025/Conference — ICLR 2025 Poster_

### Official Review · Reviewer_HReL · 2024-11-02

**Soundness:** 3
**Presentation:** 3
**Contribution:** 2
**Rating:** 6
**Confidence:** 5

**Summary:**

The paper introduces a new algorithm, LintSeq, which generates synthetic code editing sequences from existing programs. It reframes code generation as a sequence of edit generation tasks, mimicking human coding. They applied the method to create instruction + program-diff-sequence to train smaller LLMs for code synthesis. Models finetuned on these edit sequences generate more diverse programs and improve performance on benchmarks, achieving up to +20% higher scores on HumanEval pass@50 compared to baseline finetuning. Additionally, a 150M parameter model fine-tuned on these edits achieves state-of-the-art performance in code synthesis for on-device models, competing with larger models like Codex and AlphaCode.

**Strengths:**

- The idea is novel. While existing models often separate code generation and editing tasks, this paper presents a perspective to unify them into sequence of editing tasks. It's an interesting directions researchers can explore in the future.
- To prepare the training data, the authors proposed LintSeq, which refactors existing programs into sequences of static error-free code edits. This is a smart combination of program analysis and LLM.
- The pretraining experiment gives a better understanding of the effectiveness of the technique.

**Weaknesses:**

- The authors only evaluated the models on HumanEval and MBPP. While these two benchmarks are popular for most general-purpose LLMs, they are biased toward simple programs and do not cover practical coding scenarios.
- The paper used a Python-specific linter `pylint` to construct pretraining data, but it's unclear how this approach can be easily applied to other languages and be applied at scale.
- There is not enough justification for the base models used for finetuning. Why no base code LLMs?
- There is an overemphasis on pass@k when k is large. While a better pass@ larger k shows the LLM can cover a broader code distribution,  by definition, pass@1 is more practical and is what people care about more.

**Questions:**

1. See the weaknesses above.
2. Besides simple code generation tasks on HumanEval and MBPP, does this approach work for practical coding such as BigCodeBench? Also, does training on edits improve code editing tasks, say CanItEdit?
3. How does this approach handle a code file that depends on other files in a repository? In such cases, applying the linter only to the code file will raise an error.
4. For code generation tasks requiring multiple programming languages, does the same methodology still work?

Open to increase the score if the above concerns are properly addressed.

---

> ### Author Response · Authors · 2024-11-20
> **Rebuttal by Authors [1/2]**
>
> Dear Reviewer,
>
> Thank you for taking the time to provide comments on our paper. We are glad that you find the research direction explored in our paper to be interesting for future work.
>
> There are a number of important changes that we have made to our paper in response to reviewer comments and feedback, which are reflected in the updated PDF. A complete summary of these changes is provided in our general response.
>
> > The authors only evaluated the models on HumanEval and MBPP. While these two benchmarks are popular for most general-purpose LLMs, they are biased toward simple programs and do not cover practical coding scenarios.
>
> To address this limitation, we have added two additional benchmarks to our evaluations of fine-tuned Gemma 2, Phi 3, and Llama 3.1 models that cover both more practical coding scenarios (DS-1000) as well as harder program synthesis problems (CodeContests). Using these new benchmarks, we show that LintSeq also improves LM performance on tasks that are more practical and harder than those in HumanEval and MBPP. We would greatly appreciate it if you could take a look at the updated version of our paper.
>
> > The paper used a Python-specific linter pylint to construct pretraining data, but it's unclear how this approach can be easily applied to other languages and be applied at scale.
>
> The Pythonic version of LintSeq that we implemented for our experiments in this paper can be applied at scale – it requires only CPU to run and it is highly parallelizable.
>
> [There exist open-source linters like `pylint`](https://github.com/caramelomartins/awesome-linters) for just about every programming language. As a result, applying LintSeq to code written in languages other than Python can be done simply by adding a dictionary mapping file extensions (e.g. `.py`) to appropriate linter function calls. We hope to explore processing general code training corpuses (e.g. the entirety of The Stack) with LintSeq in future work.
>
> > There is not enough justification for the base models used for finetuning. Why no base code LLMs?
>
> The TinyCodeLM models that we pretrain in the paper are base code LMs. Our experiments instruction finetuning these models demonstrate that training base code LMs on instruction data refactored with LintSeq is effective for improving the quality and diversity of generations (Section 3.3.1).
>
> Our experiments with larger “off-the-shelf” LMs were designed to not only confirm that LintSeq is effective across model scales, architectures, and tokenizers, but also that it is effective for base LMs that were *not* intentionally pre-trained solely for code understanding. To our knowledge, the Gemma 2, Phi-3, and Llama 3.1 models were pre-trained with general-purpose data mixtures.
>
> We have added additional justification for this design choice to our submission (Section 3.3.2, paragraphs 1 and 2).
>
> > There is an overemphasis on pass@k when k is large. While a better pass@ larger k shows the LLM can cover a broader code distribution, by definition, pass@1 is more practical and is what people care about more.
>
> Thank you for this comment. We have updated the framing of our results throughout the paper to emphasize the Pass@1 results for LintSeqInstruct vs Instruct vs RandSeqInstruct (i.e. linter-ablated edit sequence) models.
>
> > Besides simple code generation tasks on HumanEval and MBPP, does this approach work for practical coding such as BigCodeBench? Also, does training on edits improve code editing tasks, say CanItEdit?
>
> Our updated evaluations on DS-1000 show that LintSeq models outperform baselines on practical coding tasks on pass@1. The tasks in this benchmark ask LMs to complete partially written code involving data science libraries like `matplotlib` and `numpy`.
>
> One limitation of LintSeq synthetic edit data is that it reflects insertion edits only – every edit in a LintSeq edit sequence reflects an existing line of code in the source file used to generate. It is for this reason that we evaluate our models on code generation only, rather than on general purpose code editing tasks that may require models to predict deletions. To adapt LintSeq models for general code editing settings, one could continue to train them to learn to delete and/or re-write code with RL or by mixing synthetic code edits with human edit data. This is another direction that we are very excited about for future work, and we have added additional discussion of this limitation to the “Discussion” section of our updated submission PDF.

---

> ### Author Response · Authors · 2024-11-20
> **Rebuttal by Authors [2/2]**
>
> > How does this approach handle a code file that depends on other files in a repository? In such cases, applying the linter only to the code file will raise an error.
>
> Linters like `pylint` are designed to properly understand and to correctly reflect inter-dependencies between files during error analysis. For datasets containing entire repositories, LintSeq could be run across each file from the “root” repository level, at which other files with potential dependencies would be “visible” to the linter. In this way, synthetic edit sequences could be generated at the repository level. Each edit in the training dataset would need to be modified to also contain information about the name/relative path of the file being changed.
>
> In summary, it is possible to apply LintSeq to entire repositories, though our implementation of the algorithm and the format of the individual “diffs” used during training would need to be modified to support this.
>
> ---
>
> Thank you once again for your feedback on our submission. Please let us know if you have any outstanding comments, concerns, or feedback that prevent a recommendation for acceptance.

---

> > ### Comment · Reviewer_HReL · 2024-11-21
> >
> > I appreciate the authors' response and will increase the soundness score. However, I am still concerned about the evaluation. While the inclusion of DS-1000 and CodeContests improves the coverage of the evaluation, these two benchmarks are relatively old. I am mostly interested in seeing the results on BigCodeBench [1], which covers most practical coding scenarios, as mentioned initially in my comments. Also, LiveCodeBench [2] would be a better alternative to CodeContests as it is contamination free.
> >
> > [1] https://arxiv.org/abs/2406.15877
> > [2] https://arxiv.org/abs/2403.07974

---

> > > ### Author Response · Authors · 2024-11-22
> > > **Second Rebuttal by Authors**
> > >
> > > Dear Reviewer,
> > >
> > >
> > > As requested, we have run BigCodeBench Instruct pass@1 evaluations for >1B parameter models and updated the submission PDF accordingly (please see Section 3.3.2, Figure 3, and Appendix Table 8). Our results on this benchmark align with our previous results on HumanEval, MBPP, DS-1000, and CodeContests. Namely, they suggest that finetuning on a LintSeq re-factored version of an instruction+program dataset preserves or improves pass@1 performance across models, compared to finetuning on the original data.
> > >
> > >
> > > CodeContests has been used by several very recent works to evaluate test-time compute scaling for LMs on code synthesis [1, 2, 3]. We chose to use it to strengthen our evaluations of LM performance scaling on higher pass@k as a function of FLOPs in order to align our analysis with existing works.
> > >
> > >
> > > We respectfully disagree that this benchmark is outdated, as strong LLMs like GPT-4o-2024-05-13 solve only ~20% of problems in a single attempt [1, 2]. Solutions to CodeContests problems also have longer average lengths compared to HumanEval, MBPP, DS-1000, and BigCodeBench. We believe that this property, in particular, adds depth to our analysis of edit sequence code generation.
> > >
> > >
> > > Thank you for your help in strengthening our work. We would appreciate it if you could please let us know if you have any additional concerns preventing a recommendation for acceptance.
> > >
> > > ---
> > > [1] Brown, Bradley, et al. "Large language monkeys: Scaling inference compute with repeated sampling." arXiv preprint arXiv:2407.21787 (2024).
> > >
> > > [2] Li, Jierui, et al. "CodeTree: Agent-guided Tree Search for Code Generation with Large Language Models." arXiv preprint arXiv:2411.04329 (2024).
> > >
> > > [3] Gehring, Jonas, et al. "RLEF: Grounding Code LLMs in Execution Feedback with Reinforcement Learning." arXiv preprint arXiv:2410.02089 (2024).

---

> > > > ### Comment · Reviewer_HReL · 2024-11-22
> > > >
> > > > Thanks for the additional experiment. I’ve increased my score.

---

### Official Review · Reviewer_aqpZ · 2024-11-03

**Soundness:** 3
**Presentation:** 1
**Contribution:** 2
**Rating:** 6
**Confidence:** 5

**Summary:**

This paper looks at the problem of code generation. The corpus trains LLMs to generate code from left to right. Yet, this paper proposes to generate synthetic code edits that train the model to construct code in a step-by-step fashion, while encouraging each step to preserve the syntax validity of the code. This technique is evaluated to be effective for SLMs esp. when doing inference scaling.

**Strengths:**

- This paper looks at an important domain in generative AI -- code generation.
- The overall idea of processifying code generation makes sense to me -- decomposing the code generation task into code insertions allows the model to construct the generated code with more diverse context possibilities in addition to the simple left-to-right context

**Weaknesses:**

- **Novelty & related work**: LintSeq reminds me of the code sketch paper [1] which generates code via syntax-validity-preserving edits (i.e., filling grammar holes). This paper should be discussed (and compared if applicable).
- **Technique**: One theoretical limitation of LintSeq is the lack of naturalness of synthetic edits. The data trains the model on random trajectories which might (if not always) not be meaningful.
- **Technique**: Code edits in LintSeq are *insertion-only* -- I can imagine cases deletion/rewriting can be used, such as writing a bunch of `def f(): pass` when planning the code skeleton and replacing the "pass" when real implementation starts.
- **Evaluation**: overall I feel the evaluation focuses on function-level code generation, which might look a bit contradictory to "inference scaling" -- as I'd expect to use inference scaling for solving much more challenging tasks, e.g., those in SWE-Bench.
- **Presentation**: "Open-source edit data with good coverage over the distribution of all error-free code “diffs” is nonexistent" is debatable given high-quality code-edit datasets such as EditPackFT.
- **Presentation**: The terminologies in this paper can be made more precise (detailed comments below).

----
Comments on writing:

- Fig 1: "HumanEval Coverage" and its use in many other places: maybe just explicitly say pass rate or pass @ k -- "coverage" by default means code coverage in the code generation domain and it might be confusing to refer it to pass rate.
- L065: authors should explicitly explain what "static-error-free" means -- is it program syntax error? or the error of the diff syntax, etc.
- L080 "static program verifier" and "error-free program states": how is it possible to verify an arbitrary program state "error-free" statically? :) -- Similar to the comment above, some basic concepts need to be explicitly clarified and defined.
- L084 "verifier ... is also referred to as a linter": linters and verifiers are very different concepts. linters mostly do static syntax-based checks on certain surface-level program properties. Verifiers are way more ambitious as they assure the existence/absence of program properties given all possible inputs/states (nonetheless, the meaning of verifiers has been extended to validators to assure behaviors over "certain" rather than "all" inputs/states).
- L107 "A linter is a type of standard code analysis tool that verifies the correctness of program" -- the statement is too strong to be right. Linters cannot verify program correctness but just check a few syntactical issues.

**Actionable suggestions**: for these "linters", "correctness", and "verifiers" terms, it's more precise to say "apply linters to make sure each edit can preserve the program's syntactical validity/well-formedness".

[1] LEARNING TO COMPLETE CODE WITH SKETCHES. ICML 2022.

**Questions:**

- Novelty-wise what is the paper's main difference compared to the code sketch paper?
- Code diff is more than just insertion -- why deletion and rewriting are not considered?
- Why disabling chain-of-thought data?
- Why pre-training TinyCodeLM instead of reusing existing pre-trained SLMs?
- Can you try sampling the model using min-p rather than top-k?
- What's the intuition on why LintSeq can improve pass@1? It makes sense to me it can improve pass@ big K as the approach optimizes code generation diversity but it's interesting to me that it also gets higher pass@1 (i.e., higher-quality first sample).

---

> ### Author Response · Authors · 2024-11-20
> **Rebuttal by Authors [1/2]**
>
> Dear Reviewer,
>
> We are thankful for the time that you have taken to read and review our submission.
>
> Please note that we have made a number of significant revisions to the paper during the rebuttal period, which are reflected in the updated PDF. We would appreciate it if you could also take a look at our general response, which summarizes these changes.
>
> > LintSeq reminds me of the code sketch paper [1] which generates code via syntax-validity-preserving edits (i.e., filling grammar holes). This paper should be discussed (and compared if applicable).
>
> Thank you for bringing this paper to our attention. There are several differences between the work of Guo et al. and our method. Namely, in our understanding, Guo et al.:
>
> 1. Study code completion in the context of **uncertainty**. Specifically, they investigate whether a model can be trained to decline to make predictions in parts of code where there is high uncertainty during generation.
> 2. Train their model (Grammformer) with **reinforcement learning** and a hand-engineered **reward function**. This model cannot be used for standard, left-to-right causal generation – instead, it must be run once on each partially expanded sequence during generation.
>
> In contrast, our work refactors existing programs in training corpuses across chunks whose contents and structure reflect the syntax of the programming language. We **do not study uncertainty** and we use **teacher-forced supervised learning** only to finetune off-the-shelf decoder-only models for autoregressive, edit-by-edit code generation at scale.
>
> > One theoretical limitation of LintSeq is the lack of naturalness of synthetic edits. The data trains the model on random trajectories which might (if not always) not be meaningful.
>
> Our goal in the paper is not necessarily to replace existing “natural” (i.e. human) data, but to study whether training on edits that minimally reflect the syntax of a programming language might improve LM code synthesis.
>
> > Code edits in LintSeq are insertion-only -- I can imagine cases deletion/rewriting can be used, such as writing a bunch of def f(): pass when planning the code skeleton and replacing the "pass" when real implementation starts.
>
> We agree that deletion/rewriting are just as interesting as insertion. While it is true that the edits sampled by LintSeq are insertion-only, this limitation stems from the parameter-free nature of the algorithm (i.e. every edit in a LintSeq sequence reflects an existing line of code in the source file used to generate).
>
> We are excited to explore different ways of addressing this limitation in future work. For example, it may be possible to continue to train LintSeq LMs to learn to delete and/or re-write code by using reinforcement learning and/or by mixing synthetic and human edit data (e.g. via datasets like CommitPack) during training.
>
> > Overall I feel the evaluation focuses on function-level code generation, which might look a bit contradictory to "inference scaling" -- as I'd expect to use inference scaling for solving much more challenging tasks, e.g., those in SWE-Bench.
>
> We agree and have added evaluations on two additional, challenging benchmarks (DS-1000 and CodeContests) for >1B parameter models to the paper.
>
> We would appreciate it if you could take a look at the results section (especially Figures 4 and 5) in the updated manuscript.
>
> > "Open-source edit data with good coverage over the distribution of all error-free code “diffs” is nonexistent" is debatable given high-quality code-edit datasets such as EditPackFT.
>
> Thank you for this comment, we have revised this wording of this point in the paper to be more precise. Please note that we also discuss EditPackFT in the related work section.
>
> > Comments on writing: …
>
> We have updated the paper with the suggested changes.
>
> > Why disabling chain-of-thought data?
>
> We remove CoT traces from the instruction tuning data used in our experiments purely to simplify our study. We believe that the instruction + program tuning setting is the fundamental setting where LintSeq may be applied. In principle, LintSeq can be easily combined with CoT. For example, models can be fine-tuned to “switch” between predicting code edits vs natural language traces during generation. We agree that this is an exciting direction for future work.
>
> > Why pre-training TinyCodeLM instead of reusing existing pre-trained SLMs?
>
> Our goal in this experiment is to confirm that LMs do not need to be explicitly pre-trained on code “diff” data in order for LintSeq edit sequence fine-tuning to be effective.
>
> The off-the-shelf LMs that we fine-tune in the remainder of the paper have open weights but they do not open datasets.
>
> > Can you try sampling the model using min-p rather than top-k?
>
> We have added test-time sweeps over min-p to our evaluations in the updated paper draft!

---

> ### Author Response · Authors · 2024-11-20
> **Rebuttal by Authors [2/2]**
>
> > What's the intuition on why LintSeq can improve pass@1? It makes sense to me it can improve pass@ big K as the approach optimizes code generation diversity but it's interesting to me that it also gets higher pass@1 (i.e., higher-quality first sample).
>
> There are a few possible explanations for why LintSeq may improve pass@1: (1) Synthetic edit code sequences might act as a data augmentation, since sampling several code edit sequences per program in a corpus shows models “multiple equivalent views” per example; (2) The lint-error-free nature of each sampled edits adds structure that reflects the syntax of a programming language to training data, improving the coherence / quality of the first sample.
>
> ---
> Once again, we would like to thank you for your review. We would be very grateful if you could let us know if you have any outstanding questions or additional feedback on our submission.

---

### Official Review · Reviewer_Uc3H · 2024-11-03

**Soundness:** 3
**Presentation:** 3
**Contribution:** 3
**Rating:** 8
**Confidence:** 4

**Summary:**

The paper introduces LintSeq which is a proposed method to tune LLMs for programming tasks. Data preparation consists of representing code generation as a sequence of edits. To construct the edits, there is a 2 step process broadly, first deleting lines from an existing program to create potential intermediate states, and using a linter to reject intermediate states that are syntactically incorrect or fail other kinds of static linting tests. The authors demonstrate the merits of this approach for small language models and large language models, comparing LLMs tuned with LintSeq to plain instruction-tuned models. The authors also ablate the importance of the linter-based filtering step, showing that this step is important to its success. The authors also demonstrate and explain that LintSeq-tuned models may sacrifice inference-time compute in exchange; however, in turn for higher performance on program synthesis benchmarks.

**Strengths:**

The results speak for themselves: the authors demonstrate LintSeq is capable of substantial performance improvements over instruction-tuned models. The authors are transparent that LintSeq may sacrifice inference-time compute for improvements in the efficient use of parameters and its strong program synthesis abilities. And the authors' ablation demonstrates that the use of a linter (to reject poorly formatted code edits) is important to the success of LintSeq.

In terms of intellectual merit, LintSeq is not the first work to suggest leveraging edit sequences for training language models for programing tasks. However, LintSeq is notable for getting these ideas to work at scale. To my best knowledge, much prior work may have focused on using human edits such as github commits, which is limited by the number of demonstrations available. Here: the authors introduce a procedure that is not bounded by the number of human demonstrations available, but rather the availability of a linter or even more loosely, a static-analysis based tool for checking the well-formedness of programs (e.g. tools often found in a compiler).

In terms of significance, the authors are able to demonstrate LintSeq's ability to improve over instruction tuning for a series of code LLMs. However, LintSeq also may have significance for real-world coding applications. The authors mention that commercial code assistants like Cursor deliberately do not use edits; however, the case study of LintSeq may inspire new approaches for code-editor assistants (it could also potentially have issues as well). Additionally, the set-up of LintSeq may also motivate applications for incremental generation of natural langauge responses, especially for tasks that may be intensive in reasoning.

**Weaknesses:**

The authors' work is very strong, however, there are some important points that need discussion. Overall, my largest concerns are with transparency, clarity, and integrity in presentation. I myself felt mislead multiple times while reviewing this paper, and that concerns me greatly.

1. Misleading Presentation of Models >1B parameters / Transparency + Clarity

I found figure 1 to be misleading. It compares leading LLMs on Human Eval with K=1 to LintSeq with K=50. Personally I didn't catch this until I glanced over the paper after at least multiple passes, I would fear other readers may fall into the same trap and overestimate how strong LintSeq is. In reality, LintSeq Pass@1 is can equal to or even significantly worse than an instruction-tuned variant of the same base model.

It is only in the appendix in Tables 10 and 11 where the authors clearly present LintSeq vs Instruciton-Tuning pass@1, pass@5, etc rates; although similar information is presentated in figure 4. With pass@50 LintSeq is significantly better than instruction-tuned models; whereas, it is negligible or worse with few or one/fewer samples. I believe there is no thorough discussion of this phenomenon in the main manuscript and this is an extremely important point. For example, the authors could offer an explanation that LintSeq increases the generation diversity which is highlighted in the higher pass@50. The authors must be more forthcoming about the tradeoffs of LintSeq when model size is scaled up. I urge the authors to reframe the presentation and discussion to clearly reflect this.

I would also appreciate a comparison with other top LLMs for code (e.g. LLama3 or LLama3.1; CodeLLama, DeepSeekCoder, Gpt3.5) and comparisons with equal pass@1, pass@5, pass@10 to show how these other LLMs scale with more samples. I feel very mislead by Figure 1 comparing Pass@1 for big models vs. pass@50 for LintSeq and I am left feeling such a presentation is problematic. I feel uncertain about  the motives in presenting Pass@1 vs. Pass@50 without a fair like-to-like comparison between other LLMs and putting tables 10 and 11 deep in the appendix. The authors must present a more forthcoming picture of the strengths and weaknesses of LintSeq when scaled up >1B parameters. I think the paper requires a significant revision to reflect this in language. I believe the technical contributions could already be there, but the presentation feels very misleading to me and the discussion should center much more on the tradeoffs that LintSeq introduces.

2. Framing and Explanation of LintSeq

I find the framing of LintSeq or the explanation of it is not easily clear, and even after section 2.2 where it is broken down into phase 1 and phase 2, it is still unclear to me. This led me to inspect the artifact `lintseq` in the supplementary materials which is >300 lines of code and mentions sampling weights, computing fitness, and "children." Perhaps there's something going on where in the paper "affected lines" are the indented lines of a code block. The description is not sufficiently clear, and I recommend more clarity in the presentation of the algorithm for both transparency and reader understanding. I would not call this "recursively lint" as shown in figure 2, as linting doesn't edit code, so the recursive process needs to be combined with some editing process as menitoned in section 2.2: "(2) run a linter or other
verifier on the remaining code; (3) if the deletion induced new errors, remove all affected lines; and (4) repeat steps 2 and 3 until no errors are caught by the linter." To me, the "remove all affected lines" is unclear to me and I'm not sure if all writers of the paper fully understood the procedure either. I recommend trying to formulate a more thorough explanation both in the main manuscipt and also if this is not sufficient to add an appendix section explaining the procedure in more detail. Pseudocode / pythonic pseudocode may also help to demonstrate this procedure.

2. Claims of Tiny LMs is Misleading

I'd recommend the authors walk back the claims on their Tiny LMs. I find the statement "Our 150M parameter edit sequence LM matches or outperforms code models with twice as many parameters" misleading: the 150M parameter model underperforms CodeT5+ 220M at Pass@10 HumanEval and underperforms Codex 300M at Pass@1.

3. Discussing Prior Work

The authors only mention 2 works from 2023 and 2024 on fine-tuning on edits. Some papers that came to my mind while reading this were:
https://arxiv.org/abs/1810.13337
https://arxiv.org/pdf/1905.11006 (much less related)

This makes me inclined to think that there may be more works that I have not included here that should be discussed to highlight the history of thinking on this issue.

4.  Choice of Language

The term `static program verifier` makes a loose use of the word "verify." I recommend using a different word from "verify" to not conflate it with [formal verification](https://en.wikipedia.org/wiki/Formal_verification) of programs.

Also in the abstract "small LLMs" is a contradiction of sorts "small Large Language Models."

**Questions:**

Generally, this is a re-hashing of the weaknesses

1. Could you please describe the algorithm for LintSeq more clearly, and I'd strongly also recommend updating the paper to reflect a more clear explanation (e.g. PseudoCode or a more lengthy explanation in natural language in an appendix section).

2. Did I misrepresent or misunderstand your claims? Please feel free to disagree if I have missed something.

3. Could you provide comparisons to other commercial LLMs (e.g. Gpt4o, the updated Sonnet 3.5, Gpt4o-mini, Haiku 3) and open LLMs (e.g. the most recent LLama3-Instruct 70b, Llama3-Instruct 8b, DeepSeekCoder 7b instruct, DeepSeekCoder33b Instruct) on MBPP and HumanEval with higher samples, and demonstrating pass @N as you did in tables 10 and 11.

4. For the models in tables 10 and 11, is it possible to also compare lintseq best@50 to instruction-tuned models best@50, except with Top p = 1.0 and temperatures [1.0, 1.1, 1.2]? This is important to also establish if the increase in diversity of lintseq cannot be matched by modulating the temperature for sampling.

5. Are there other ways to calculate estimated FLOPs as you have? Would it be possible to also tell me the wall clock time of running inference using the instruction-tuned models and compare that to the wall clock time of using the LintSeq models (e.g. both 14b models compared) just for more transparency.

---

> ### Author Response · Authors · 2024-11-20
> **Rebuttal by Authors [1/2]**
>
> Dear Reviewer,
>
> We are sincerely grateful for your thorough feedback on our paper. We are glad that you thought our work was very strong, and we agree that the results of our approach speak for themselves. We apologize that you felt misled by our presentation of some of these results. This was not our intention.
>
> Please note we have made a number of changes to the paper during this rebuttal period, as reflected in the revised PDF. These changes are summarized both in our general response and in our responses to your comments below – we believe they substantially improve the clarity and strength of our work.
>
> > I found figure 1 to be misleading. It compares leading LLMs on Human Eval with K=1 to LintSeq with K=50.
>
> We apologize that you found the previous version of Figure 1 to be misleading. Our intention had been for this figure to serve as a “teaser” comparing the performance vs total test-time FLOPs used by LintSeqInstruct and Instruct LMs at pass@50. We only included pass@1 from existing, leading, and much larger LMs with corresponding total test-time FLOP estimates to contextualize scores and compute costs.
>
> In the updated version of the paper, we have removed the comparison of LintSeq models to leading LLMs from Figure 1. The new version of the figure shows the effectiveness of our method for improving test-time scaling curves (i.e. pass@k vs. total test-time FLOPs to get k samples) on instruction-tuned model variants on a fixed dataset.
>
> > In reality, LintSeq Pass@1 is can equal to or even significantly worse than an instruction-tuned variant of the same base model.…I believe there is no thorough discussion of this phenomenon in the main manuscript and this is an extremely important point.
>
> We respectfully disagree. As described above, we have revised the paper to (1) include estimates of standard error for all computed benchmark scores and (2) reflect more comprehensive evaluations for >1B models. In our updated results, LintSeq statistically significantly improves performance across LMs on all but one of two of the benchmark metrics tested (HumanEval pass@1 and CodeContests pass@1). Notably, even on these metrics, the two least performant LintSeq instruction-tuned models (Llama 3.1 8B and Phi-3 14B) still achieve performance that is statistically indistinguishable from the baseline. On higher pass@k, we see strong improvement on all models and on all benchmarks.
>
> > I would also appreciate a comparison with other top LLMs for code (e.g. LLama3 or LLama3.1; CodeLLama, DeepSeekCoder, Gpt3.5) and comparisons with equal pass@1, pass@5, pass@10 to show how these other LLMs scale with more samples. I feel very mislead by Figure 1 comparing Pass@1 for big models vs. pass@50 for LintSeq and I am left feeling such a presentation is problematic.
>
> Again, we apologize that you felt misled. The goal of our experiments was not necessarily to develop the strongest code model possible, but rather to fairly and comprehensively probe whether tuning LMs to generate code edit-by-edit by preprocessing data might improve performance over standard tuning on a fixed dataset.
>
> We would also like to clarify that there are a few other prominent differences between LintSeq models vs the top LLMs for code that you have mentioned (e.g. CodeLlama, DeepSeekCoder, GPT3.5). First, these top LLMs are very likely to have been trained/tuned on code data with chain-of-thought (CoT) traces. In contrast, our goal in this paper was to conduct the simplest possible comparison between LintSeq vs baseline models, and so we did not tune with any CoT data. Second, the hyperparameters that were used during fine-tuning for top code LLMs were likely tuned to improve benchmark pass@1 as much as possible. We did no such tuning in this paper, though we did do some tuning of learning rates in the baseline instruction tuning setting for each model.
>
> For these reasons, we have substantially reframed the discussion of our results in the new draft to center on only comparing LintSeqInstruct to Instruct models and ablations throughout the text, removing direct comparisons to external code LLMs.
>
> We remain open to adding comparisons between LintSeqInstruct and Instruct models vs top LLMs to the Appendix in a future draft, if you believe it would help to improve the reader’s understanding of the method.
>
> > I feel uncertain about the motives…putting tables 10 and 11 deep in the appendix
>
> We did not intend to “hide” these tables – the data they contain was present in Figure 4 of the submitted manuscript. In the updated draft, we have moved them to the first few pages of the Appendix and we have prominently highlighted pass@1 scores of LintSeq vs baseline instruction tuned models in Figure 3.

---

> ### Author Response · Authors · 2024-11-20
> **Rebuttal by Authors [2/2]**
>
> > I find the framing of LintSeq or the explanation of it is not easily clear, and even after section 2.2 where it is broken down into phase 1 and phase 2, it is still unclear to me.
>
> Apologies, we have added pseudocode in Figure 2 of the methods section and revised the description of the algorithm. We hope you find the updated text to be clearer.
>
> > This led me to inspect the artifact lintseq in the supplementary materials which is >300 lines of code and mentions sampling weights, computing fitness, and "children."
>
> We apologize that the implementation of LintSeq for Python in our submission was not adequately cleaned up and documented. We have updated the code with additional documentation.
>
> Our submitted implementation of the algorithm supports (1) a few additional Python-specific heuristics designed to reduce the number of linter calls made by LintSeq and (2) extra hyperparameters inspired by evolutionary search that can be used to extend the algorithm. For example, these parameters allow a user to up-sample a particular type of edit (e.g. indented blocks) so that such edits occur earlier in training sequences. Similarly, the parameter “children_per_round” allows a user to modulate the number of edits expanded from each sampled intermediate program state beyond 1 at every round of LintSeq.
>
> These additional heuristics and hyperparameters were **not used in any of our experiments**. They should have been clearly documented in the code (and now are), and we once again sincerely apologize for the confusion.
>
> > I'd recommend the authors walk back the claims on their Tiny LMs.
>
> We have adjusted this in the updated paper.
>
> > Some papers that came to my mind while reading this were: https://arxiv.org/abs/1810.13337 https://arxiv.org/pdf/1905.11006 (much less related)
>
> Thank you for bringing these papers to our attention. We have added them to our related work section, which we have generally updated to reflect earlier work fine-tuning on edits.
>
> > I recommend using a different word from "verify" to not conflate it with formal verification of programs.
>
> We have updated this in the paper.
>
> > is it possible to also compare lintseq best@50 to instruction-tuned models best@50, except with Top p = 1.0 and temperatures [1.0, 1.1, 1.2]
>
> We ran this expanded evaluation and updated our reported results. Overall, we found that LintSeq models continue to strongly outperform baselines in diversity even after modulating sampling temperature.
>
> > Are there other ways to calculate estimated FLOPs as you have? Would it be possible to also tell me the wall clock time of running inference using the instruction-tuned models and compare that to the wall clock time of using the LintSeq models (e.g. both 14b models compared) just for more transparency.
>
> To our knowledge, estimating total FLOPs with tokens is the standard way to compare total test time costs between models. Our evaluations were conducted across a range of hardware (i.e. nodes with different numbers of available GPUs), so we cannot directly compare wall clock time. In an effort to add transparency to the paper, we have added token-based total FLOPs comparisons to all figures visualizing the benchmark scores of models during repeated sampling (Figure 1(right) and Figure 5).
>
> ---
> Thank you once again for your very thoughtful review – it greatly helped us improve our paper. We would appreciate it if you could let us know if you have any outstanding questions or concerns.

---

> > ### Comment · Reviewer_Uc3H · 2024-12-02
> >
> > Thank you for the revisions. The paper is substantially improved: the pseudocode addition, clearer method description, improved figure for the data preparation algorithm, and reframed comparisons provide a much better presentation of the work's contribution.
> >
> > I must note that the shifts in framing and positioning between the original submission and revision (particularly the claims on TinyLMs, regarding comparisons to top LLMs, the clarity of pass@1 results) suggest the initial presentation wasn't merely unclear, but was actively positioning the work more strongly than warranted. While these issues are now addressed, I find it difficult to accept the suggestion that this wasn't the original intent - it feels disingenuous to me. It would have been more appropriate to acknowledge this repositioning directly rather than suggest it wasn't the original intent. While I cannot read minds or truly ascertain intent, I would be disingenuous to myself if I didn't maintain this point.
> >
> > Given the technical merit of the work and the thorough improvements made in the revision, out of context and as a paper alone, I feel fairly confident I would recommend acceptance for this work. The revised presentation appropriately contextualizes the results.

---

> > > ### Author Response · Authors · 2024-12-03
> > > **Second Rebuttal by Authors**
> > >
> > > Dear Reviewer,
> > >
> > >
> > > We stand by our initial positioning of the results, and we hope to assure you that we did not intend to mislead the reader about the strength of our method in any draft of our paper.
> > >
> > >
> > > For example, we provided a discussion of the mixed nature of pass@1 results for >1B parameter models in Section 3.3.2 of the main text in our first draft.
> > >
> > >
> > > > At pass@1, however, our results are slightly more mixed than in Section 3.3.1. For Phi-3 models, we observe either no difference or a decrease in score between each pair of model-data variants… **(Lines 369-373, Submission on 10/01/2024)**
> > >
> > >
> > > Please note that the effect described above was caused by a bug in our evaluation code, which we found and eliminated during the Rebuttal Period  (“General Response,” 11/19/2024). Re-running all of our model evaluations confirmed that LintSeqInstruct Pass@1 is either comparable to or better compared to Instruct Pass@1 for all tested models and across four different benchmarks, as indicated in our updated submission PDF.
> > >
> > >
> > > Furthermore, our previous claim about the performance of TinyCodeLMs, namely,
> > >
> > >
> > > > our smaller edit sequence-finetuned model is particularly strong for its size, roughly matching or out-performing models with twice as many parameters including the 300M parameter version of Codex and the 302M-parameter version of AlphaCode **(Lines 369-373, Submission on 10/01/2024)**
> > >
> > >
> > > was and continues to be substantiated by our results. The 150M parameter version of TinyCodeLM-LintSeqInstruct achieves a pass@1 of 12.8 on HumanEval, while AlphaCode 302M and Codex 300M have reported pass@1 scores of 11.6 and 13.2, respectively. On HumanEval pass@10, TinyCodeLM-LintSeqInstruct 150M achieves a score of 20.6 while AlphaCode 302M and Codex 300M have reported scores 18.8 and 20.4, respectively.
> > >
> > >
> > > Finally, we did our best to clearly indicate that the previous version of Figure 1 reflected performance on HumanEval as a function of total inference-time FLOPs from different models, rather than performance at fixed pass@k.
> > >
> > >
> > > We are very grateful for your time, expertise, and favorable re-assessment of our work. Your feedback was critical for improving our paper, and we hope that our work can be judged for its contribution to the community.

---

### Official Review · Reviewer_8Cje · 2024-11-04

**Soundness:** 2
**Presentation:** 3
**Contribution:** 3
**Rating:** 6
**Confidence:** 3

**Summary:**

This paper proposes LintSeq, which is a synthetic data generation algorithm that generates edit sequences from raw code to construct training datasets for LLMs. Specifically, to let LLMs learn to write code by editing existing programs like human developers, LintSeq first refactors existing code into a sequence of code edits, where each edit is an error-free insertion, and then outputs edit sequences as text strings consisting of consecutive program diffs. Evaluation shows that, after training LLMs on the synthetic edit sequences, the inference-time scaling performance of LLMs on HumanEval and MBPP improves significantly.

**Strengths:**

- The paper explores an interesting and important research direction.
- The idea of constructing fine-tuning datasets that mimics human developers’ behavior by synthesizing edit sequences is novel.

**Weaknesses:**

- The synthesized edit sequences may not be reasonable. While LintSeq ensures that the program at each step of the edit sequence does not contain any static error, it does not necessarily guarantee that this edit sequence is reasonable because a valid edit sequence should reflect the reasoning/thinking process of human developers, which is not guaranteed by LintSeq. Furthermore, edit sequences synthesized by LintSeq only contain insertions, which is also different from real-world edit sequences.
- The comparison between “LintSeqInstruct” and “Instruct” may not be fair enough. While authors have controlled the number of optimizer steps the same, it is also important to make sure that “LintSeqInstruct” and “Instruct” share the same valid training tokens (i.e., the number of tokens in the training sequences that are not padding tokens). Although sharing the same number of optimizer steps, it is still possible that each training sequence in “LintSeqInstruct” is much longer than that in “Instruct” and thus more training tokens are provided for “LintSeqInstruct”.
- Evaluation is not comprehensive enough. Apart from HumanEval(+) and MBPP(+), more benchmarks should be included to further demonstrate the effectiveness of “LintSeqInstruct”, such as DS-1000, APPS, and LiveCodeBench.

**Questions:**

- What are the computational resources required to synthesize edit sequences using LintSeq? If it is cost friendly, will it also be able to improve pre-training performance?

---

> ### Author Response · Authors · 2024-11-20
> **Rebuttal by Authors**
>
> Dear Reviewer,
>
> Thank you for your thoughtful feedback on our submission. We are glad that you thought our paper explores an interesting and important research direction.
>
> Please note we have made a number of changes to the paper during this rebuttal period that we believe improve the strength of our work. These changes are summarized in our general response, and are reflected in the updated PDF.
>
>
> > The comparison between “LintSeqInstruct” and “Instruct” may not be fair enough. While authors have controlled the number of optimizer steps the same, it is also important to make sure that “LintSeqInstruct” and “Instruct” share the same valid training tokens (i.e., the number of tokens in the training sequences that are not padding tokens).
>
> We perform an experiment that controls for total training token count in Section 3.4 of the paper, where we ablate the linter from LintSeq to compare LintSeqInstruct TinyCodeLM models against LMs that are instruction tuned on the same quantity of entirely randomly sampled insertions, i.e. “RandSeqInstruct.” In this experiment, both sets of models are trained for the same number of optimizer steps *and* on 18e6 valid training tokens. Even so, RandSeqInstruct LMs underperform their LintSeqInstruct counterparts by 24-30% on relative MBPP and HumanEval pass@1. This result suggests that performance gains from tuning on LintSeq vs standard data cannot simply be attributed to extra training tokens.
>
> We have updated our paper to make an explicit note of the token-count equivalence of the LintSeqInstruct and RandSeqInstruct datasets in Section 3.4.
>
> > Evaluation is not comprehensive enough. Apart from HumanEval(+) and MBPP(+), more benchmarks should be included to further demonstrate the effectiveness of “LintSeqInstruct”, such as DS-1000, APPS, and LiveCodeBench.
>
> We agree and have updated our paper with additional evaluations on two new benchmarks: DS-1000 and CodeContests.
>
> > What are the computational resources required to synthesize edit sequences using LintSeq? If it is cost friendly, will it also be able to improve pre-training performance?
>
> LintSeq is highly parallelizable and requires CPU-only to run. We were able to generate the full training dataset used in our experiments in under 10 minutes on 64 CPU cores.
>
> We do believe that it can also be used to improve pre-training performance, and we are really excited to explore this direction in future work.
>
> > The synthesized edit sequences may not be reasonable. While LintSeq ensures that the program at each step of the edit sequence does not contain any static error, it does not necessarily guarantee that this edit sequence is reasonable because a valid edit sequence should reflect the reasoning/thinking process of human developers, which is not guaranteed by LintSeq.
>
> We are somewhat confused by this comment.
>
> Open-source and sequential data that reflects the process by which human developers write code is scarce. The intention behind LintSeq is not to replace this existing human data, but to explore whether generating and training language models on synthetic edits might improve LM performance on code generation tasks. LintSeq can be used to generate pre-training scale synthetic edit sequences.
>
> The contents of insertion edits sampled with LintSeq are informed by the syntax of the programming language that code is written in. Given a target program and a linter for the corresponding programming language, LintSeq edits *do* represent the minimal granularity at which a developer could make syntactically correct (i.e. linter-error-free) insertion edits to a file while writing this program. In this sense, while LintSeq edits may not reflect the reasoning/thinking used by a developer who wrote a particular file, they *do* reflect more abstractions of a programming language compared to “raw” code.
>
> ---
>
> Thank you once again for the time you have taken to review our paper, and please let us know if you have any outstanding concerns that stand between us and a recommendation for acceptance.

---

> > ### Comment · Reviewer_8Cje · 2024-12-02
> >
> > Thank you for your additional results and discussion and I have raised my score to 6.

---

### Author Response · Authors · 2024-11-20
**General Response**

We thank all of the reviewers for their thorough, insightful & constructive comments. We are glad that you found the research direction explored in our submission to be interesting and important (8Cje, Uc3H, aqpZ). While there are many existing works that study training LMs on edit data, we agree that – at least to our knowledge – LintSeq is the first method for “getting these ideas to work at scale” (Uc3H).

Several reviewers expressed concerns about the comprehensiveness of our evaluations and the presentation of our results. In response to this feedback, we have updated our submission with a number of important adjustments to the paper that we believe significantly improve the work.

These changes are summarized below.


1\. **More comprehensive evaluations of every tested model (additional hyperparameters and new benchmarks for >1B parameter models) [8Cje, Uc3H, HReL, aqpZ]**

* 1.1\.  To compare LintSeq models against baselines on harder and more practical coding tasks, we added evaluations on **CodeContests** [1] and **DS-1000** [2] for all >1B parameter models to the paper.
* 1.2\. Additionally, we added many new sampling hyperparameters to our HumanEval and MBPP(+) evaluation sweeps (top-p [0.95, 1.0], temperature [1.0, 1.1, 1.2], min-p [0, 0.5]). As before, we report the best pass@k achieved by each model after tuning on these extended sampling hyperparameters sets across each benchmark.

2\.  **Reporting of standard errors for all computed performance metrics on HumanEval, MBPP, and CodeContests** (computed following guidelines from [3] for correctly computing error bars on “pass@k” scores)

3\. **Demonstration that LintSeqInstruct Pass@1 is either comparable or better compared to Instruct Pass@1 for all tested models and benchmarks**
* 3.1\. Aside from running many additional evaluations, we also identified and eliminated a bug in one of our HumanEval evaluation launch scripts – this bug had prevented correct reporting of HumanEval results for LintSeqInstruct Phi-3 14B in the previous draft of the paper.

4\.  **Significant revision to our presentation of the results [Uc3H, aqpZ, HReL]**

* 4.1\. Adjustment to Figure 1 and all figures in the results section.
* 4.2\.  Merging of Tables 1 and 2 (i.e. all HumanEval and MBPP scores for TinyCodeLM) into a single table for more transparent comparison of TinyCodeLM performance and adjustment to language.
* 4.3\. Emphasized comparison of the more practical pass@1 setting for LintSeqInstruct vs Instruct models throughout the paper.
* 4.4\. More transparent presentation of the tradeoffs between LintSeqInstruct vs Instruct models at test-time.

5 \. **Revised framing and explanation of the LintSeq algorithm [Uc3H]**
* 5.1\. Addition of pseudo-code to Figure 2 and clear definitions of important terms.

6\.  **More detailed discussion of prior work finetuning LMs on edit data [Uc3H, aqpZ]**

7\.  **Revised and more precise language discussing linters as tools for checking code for syntactical issues, rather than as formal verifiers, throughout the paper [Uc3H, aqpZ]**

---
[1] Li, Yujia, et al. "Competition-level code generation with alphacode." Science 378.6624 (2022): 1092-1097.

[2] Lai, Yuhang, et al. "DS-1000: A natural and reliable benchmark for data science code generation." International Conference on Machine Learning. PMLR, 2023.

[3] Miller, Evan. "Adding Error Bars to Evals: A Statistical Approach to Language Model Evaluations." arXiv preprint arXiv:2411.00640 (2024).

---

### Author Response · Authors · 2024-11-22
**General Response II**

Dear Reviewers,


We have now added **additional pass@1 evaluations on BigCodeBench [1] for all >1B parameter models** to our submission, as suggested by Reviewer HReL. Our findings on this fifth benchmark echo our existing results on HumanEval, MBPP, CodeContests, and DS-1000: **as on the previously evaluated benchmarks, we show that tuning on a LintSeq-refactored dataset of instruction+program pairs preserves or improves pass@1** performance on BigCodeBench problems compared to tuning on the original version of the dataset.


The new results are discussed and/or shown in Section 3.3.2, Figure 3, and Table 8 of the updated PDF.  To align with our existing evaluations, we use default “Instruct”-style prompts to evaluate all models. As in [1], we generate completions with greedy decoding.



As the rebuttal deadline is approaching soon, we kindly ask whether you could please let us know if you have any further outstanding questions or concerns that prevent a recommendation for acceptance.

---
[1] Zhuo, Terry Yue, et al. "BigCodeBench: Benchmarking code generation with diverse function calls and complex instructions." arXiv preprint arXiv:2406.15877 (2024).



---

**Update 11/24/2024**: We found and fixed an error in Figure 3. The submission PDF has been updated accordingly.

---

### Author Response · Authors · 2024-11-25

Dear Reviewers,

As the review period ends on November 26th AoE, we would greatly appreciate it if you could please take a look at our updated submission and responses to your reviews, if you have not yet. We have made an effort to individually address each of your concerns.


We are very grateful for your time and your help in improving our paper.

---

### Author Response · Authors · 2024-12-02

Dear Reviewers 8Cje, Uc3H, and aqpZ,

We would like to kindly remind you that today (December 2nd, EoD AoE) is the last day for reviewers to provide feedback. We can respond to any further questions or concerns until tomorrow.

We have addressed your comments in detail, added new evaluations, and made significant updates to our submission. If there is anything else you would like us to clarify, please let us know.

Thank you for your time,

The Authors

---

### Meta-Review · Area_Chair_j3ES · 2024-12-19

**Metareview:**

This paper presents LintSeq, an algorithm that refactors existing code into sequences of structured edits for training language models. The key strengths include: (1) Novel approach to generate synthetic edit sequences that preserve syntax validity using linters, (2) Comprehensive evaluations showing improved performance across multiple benchmarks, especially for higher pass@k values, and (3) Effective application to both small and large language models. Main weaknesses include initial unclear presentation of pass@1 vs pass@k results and heavy focus on insertion-only edits. Overall, the paper makes a valuable technical contribution by demonstrating how edit-based training can improve code generation, particularly for scaling inference compute, warranting acceptance despite some limitations.

**Additional Comments On Reviewer Discussion:**

During rebuttal, authors addressed key concerns by: adding BigCodeBench and DS-1000 evaluations, clarifying pass@1 performance, improving algorithm explanation with pseudocode, and refining claims about TinyLMs. All reviewers acknowledged the improvements made through rebuttal. Remaining minor concerns about synthetic edit naturalness and insertion-only limitations were adequately addressed by authors' responses explaining the design choices.

---

### Decision · Program_Chairs · 2025-01-22

Accept (Poster)